# Complete Cyclic Subtask Graphs for Tool-Using LLM Agents: Flexibility, Cost, and Bottlenecks in Long-Horizon Workflows

## Abstract

Long-horizon tool-using tasks sometimes benefit from revisiting earlier subtasks, but explicit revisitation also adds routing, coordination, and token cost. We study complete cyclic subtask graphs for large language model (LLM) agents: a workflow controller in which executable subtasks are fully connected and a unified state-analysis-and-routing agent selects transitions from natural-language criteria. We evaluate task-specific (Spec-Cyc) and benchmark-generic (Gen-Cyc) cyclic graphs on TextCraft, ALFWorld, and Finance-Agent against ReAct and dependency-directed workflows. Our main dependency-directed controller is DepDAG, which permits same-subtask retry while preserving forward dependency constraints. The evaluated case studies suggest three workflow signatures rather than a universal architecture ranking. TextCraft behaves like a prerequisite-chain setting, where cyclic routing often adds overhead. ALFWorld behaves like a partially observable recovery setting, where explicit revisitation improves exploration and success. Finance-Agent behaves like an open-ended evidence-synthesis setting, where workflow control alone is insufficient without stronger retrieval, grounding, and verification mechanisms. We add a qualified workflow-signature matrix, fault-injection robustness analysis, token-cost accounting, graph-stability reporting, transition-audit checks for DepDAG, and failure-mode structure for trajectory analysis. Overall, complete cyclic subtask graphs are best understood as a diagnostic workflow-control tool: they expose when flexible backtracking is worth its cost and when simpler, locally retrying, or sparsified controllers are preferable.

## 1 Introduction

Large language models (LLMs) are increasingly deployed as tool-using agents that interact with external environments such as the web, APIs, simulators, and embodied text worlds. Recent systems range from ReAct-style closed-loop reasoning/action to explicit planner-executor decompositions and state- or graph-based controllers Yao et al. (2022); Erdogan et al. (2025); Wu et al. (2024); Zhuge et al. (2024). In these long-horizon settings, errors compound across multi-step trajectories and partial progress must often be preserved, motivating controllers that can verify state, replan, and recover rather than only continue forward. Recent work similarly treats LLM agents as long-horizon learning systems whose reliability depends on roles, execution, external knowledge, reflection, and workflow planning rather than on prompting alone (Chen et al., 2026). The same design pressure appears in analyses of LLM reasoning failures and efficient reasoning, which emphasize both robustness failures and the token/latency cost of longer reasoning traces (Song et al., 2026; Sui et al., 2025).

Despite this shift, revisitation is rarely treated as a first-class capability: "going back" is often implemented through retries, edited step lists, or ad hoc recovery states rather than through an explicit workflow representation. This distinction matters because control-flow structure shapes both behavior (e.g., how readily an agent can backtrack or explore) and diagnosis (e.g., whether repeated behavior reflects productive recovery or unproductive looping).

---

*Large-language-model assistance was used for language editing. All ideas, claims, experiments, results, and final manuscript decisions are the authors' responsibility.

We study complete cyclic task graphs as an explicit workflow representation for tool-using agents. The task is decomposed into subtask nodes, and a unified analyzer+router selects transitions by evaluating natural-language edge criteria against a rolling trajectory state (Sec. 3). We instantiate Spec-Cyc (task-specific graphs) and Gen-Cyc (benchmark-generic graphs reused across instances) to test whether cyclic revisitation acts as a transferable control policy rather than a per-task artifact. Our goal is not to claim complete connectivity is universally optimal; rather, we use it as a lens to understand when revisitation helps and when it degenerates into thrashing.

This work is best viewed as a controlled study of workflow flexibility versus coordination cost rather than as a pure leaderboard comparison. Our complete cyclic graph is a deliberately permissive multi-agent regime: because every subtask can transition to every other, the controller is not confined to a predefined trajectory, and revisitation, recovery, and exploration are always structurally available. The central scientific question is therefore when that added control-flow freedom buys meaningful recovery relative to a simpler ReAct agent, and when it mostly adds coordination and token overhead. This framing also motivates our model choices. We intentionally center the controlled comparisons on `gpt-4o-mini`, while also comparing against stronger `gpt-5-mini` settings, because once stronger single-agent or high-tier multi-agent configurations approach saturation, there is far less headroom to observe the recovery effects of orchestration itself.

We define the graph at the level of executable subtasks of the task itself. Each node corresponds to an action-producing subproblem, and the router selects among those subtask-specific action spaces using natural-language transition criteria. Because every subtask can route to every other subtask, nothing is pruned a priori. We use this deliberately maximally flexible design as an experimental lens: it lets us study when unrestricted revisitation improves recovery or exploration, when it induces thrashing or routing hallucinations, and how those effects interact with router quality, tool exposure, robustness perturbations, and token cost.

We evaluate on three structurally distinct long-horizon benchmarks, TextCraft, ALFWorld, and Finance-Agent, against ReAct and dependency-directed workflows. The main dependency-directed workflow is DepDAG: each subtask may either be retried locally, advance along the dependency graph, or terminate. This preserves the acyclic structure of inter-subtask movement while allowing local repair. We further ablate orchestration tiering, tool access (generalist full-tool executors vs. tool-restricted specialists), optional $n$-shot successful-trajectory conditioning (akin to workflow/routine induction and reuse for long-horizon agents Wang et al. (2025b)), and robustness under control-flow perturbations.

The three benchmarks stress different failure modes and should be read as case studies rather than exhaustive representatives of broad task classes. TextCraft primarily rewards correct execution of a prerequisite chain once that chain is identified, so extra routing flexibility can become coordination overhead. ALFWorld is partially observable and interaction-heavy: agents must search, recover from mis-localization, and revisit earlier subtasks after failed or misleading actions, so explicit backtracking is often useful. Finance-Agent is open-world and evidence-heavy; here low success across controllers suggests that retrieval, grounding, and evidence synthesis remain limiting factors. This difference in structure helps explain why cyclic control is most compelling in ALFWorld, mixed in Finance-Agent, and often less efficient than a single ReAct agent in TextCraft, consistent with recent findings that multi-agent coordination can underperform single-agent execution on sequential, dependency-heavy planning tasks such as PlanCraft Kim et al. (2025).

**Contributions.**

- We operationalize complete cyclic subtask graphs as a controlled upper-flexibility condition for tool-using LLM workflows, with executable subtasks fully connected and transitions selected from natural-language criteria.

- We compare task-specific and benchmark-generic cyclic control against ReAct and DepDAG, which permits same-subtask retry while preserving forward dependency constraints.

- We present a qualified workflow-signature matrix that summarizes hypotheses suggested by the three evaluated benchmark case studies rather than a universal architecture selector.

- We report success, efficiency, revisitation, transition, exploration, token-cost, failure-mode, and fault-injection analyses to measure when unrestricted revisitation helps and when it mainly adds cost.

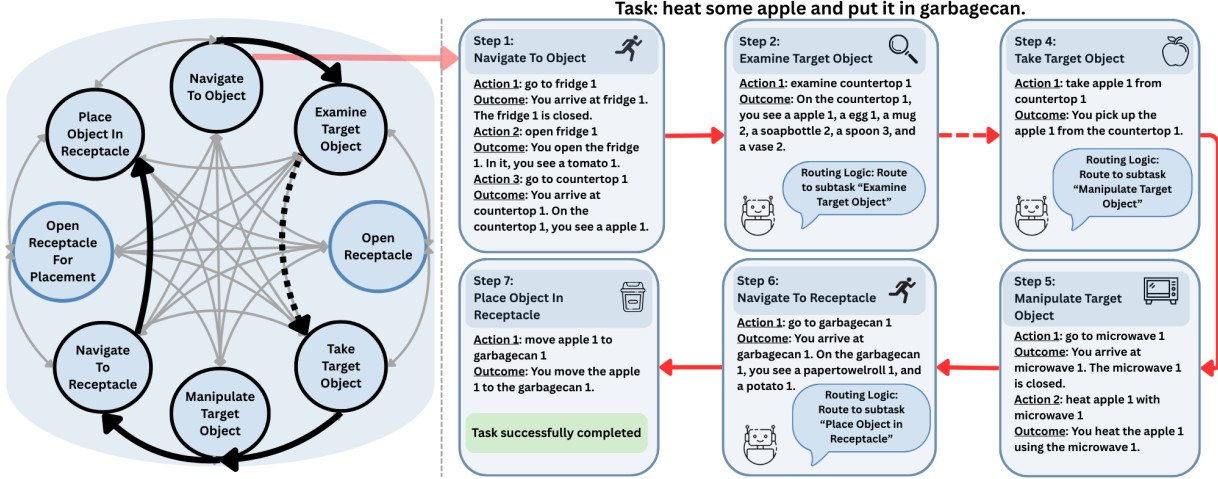

Figure 1: Diagram showing a complete cyclic graph from a real ALFWorld task experiment with executions and routing transitions (the routing criteria and some subtasks and routing logics are omitted for visual purposes). Dashed lines indicate that other subtasks were visited within the trajectory.

Taken together, our results suggest that making revisitation explicit through cyclic workflow control can be a useful and interpretable design point for long-horizon tool-using agents, while clarifying that performance hinges on orchestration quality and tool exposure, and that unrestricted revisitation can either enable recovery or amplify thrashing.

## 2 Related Work

Agentic LLM systems increasingly close the loop between reasoning, tool use, and external feedback. ReAct interleaves reasoning and actions during environment interaction Yao et al. (2022), Reflexion adds self-evaluation and memory across attempts Shinn et al. (2023), and related work studies alternative action interfaces such as executable code actions Wang et al. (2024). A broader line separates planning from execution for long-horizon tasks: Plan-and-Act studies planner–executor decomposition in web navigation Erdogan et al. (2025), ADaPT recursively decomposes subtasks and refines plans online when execution stalls Prasad et al. (2024), CoPE studies concurrent planning and acting under commitment constraints Coles et al. (2025), OPEx analyzes embodied agents by swapping Observer/Planner/Executor components Shi et al. (2024), and ReWOO decouples reasoning traces from tool observations for more efficient tool use Xu et al. (2023). Our work shares this closed-loop view, but focuses on a different design variable: the workflow representation itself. We ask what changes when the controller is an explicit complete cyclic subtask graph whose revisitation behavior can be directly measured.

Several recent frameworks make workflow control more explicit. AIME coordinates iterative planning and dispatch through centralized progress management Shi et al. (2025a), and AgentOrchestra maintains plan state through step creation, update, and deletion operations Zhang et al. (2025e). State-machine and graph-based approaches make the controller even more explicit: StateFlow represents task solving as a state machine with rule- or LLM-governed transitions Wu et al. (2024), MetaAgent uses a finite-state-machine backbone for constructing multi-agent systems Zhang et al. (2025f), and GPTSwarm frames language-agent systems as optimizable graphs Zhuge et al. (2024). Recent surveys of data-science agents, GUI agents, and test-time-compute search similarly emphasize the separation of planning, execution, reflection, external evidence, and routing policies (Chen et al., 2026; Zhang et al., 2025a; Li, 2025).

These systems establish that graph-structured workflows, explicit states, and cyclic transitions are not themselves new. Our contribution is an empirical intervention: we fix the executable-subtask graph to complete connectivity, thereby isolating unrestricted cross-subtask revisitation, and compare its recovery and cost against implicit ReAct control and dependency-directed local retry. The complete graph is therefore a

diagnostic upper-flexibility condition, not a proposed general-purpose state-machine formalism or deployment-optimal topology.

A large body of work also studies how multiple LLM agents should be organized, communicate, and coordinate. Orchestration frameworks can dispatch specialized agents and integrate tool calls through centralized schedulers Song et al. (2025), while other systems treat multi-agent communication topology as a first-class design object, including G-Designer, MaAS, MAS-GPT, and large-scale DAG collaboration networks such as MacNet Zhang et al. (2024; 2025b); Ye et al. (2025); Qian et al. (2024). Multi-agent debate and structured interaction are likewise used to improve reasoning through aggregation or competition Smit et al. (2024); Chen et al. (2024); Klein et al. (2024). These works show that coordination structure matters, but they typically focus on agent-to-agent communication or agent-graph organization. In contrast, our graph nodes are executable subtasks of the task itself, and the complete graph is used as a diagnostic workflow controller for studying recovery, exploration, and redundant revisitation.

Tool selection and specialization are another practical constraint in tool-using agents. ToolLLM/ToolBench, ToolRet, Meta-Tool/Meta-Bench, and Gorilla study API retrieval and open-world tool use, showing that narrowing candidate tools can reduce hallucination but introduces a retrieval dependency that can affect downstream success Qin et al. (2024); Shi et al. (2025b); Qin et al. (2025); Patil et al. (2024). Other systems argue for role or tool specialization to reduce interference between high-level reasoning and low-level execution Wang et al. (2025a); Song et al. (2024). Our experiments directly ablate this issue by comparing generalist full-tool executors against tool-restricted specialists within the same cyclic controller, allowing us to separate tool-selection effects from coordination and hand-off effects.

Finally, robustness and debugging are increasingly central in multi-agent systems, including fault-injected collaboration studies, failure attribution, and perturbation-based testing Huang et al. (2025); Zhang et al. (2025d;c); Ma et al. (2024). Our robustness study is aligned with this stress-testing perspective but targets control flow specifically: we inject random subtask redirections to test whether cyclic controllers can recover under disrupted routing, and we pair success rates with behavioral measures of revisitation and exploration.

## 3 Methodology

Given an objective $O$ and tool/environment interface $\mathcal{U}$, we execute via a complete directed subtask graph in which each node is handled by a tool-using executor and a separate analyzer+router selects the next node by evaluating natural-language transition criteria on outgoing edges, making backtracking/revisitation always available.

### 3.1 Problem setting and time scales

We consider episodic interaction via tools $\mathcal{U}$. Let $k$ index tool calls and let $m$ index subtask segments: within segment $m$, executor $\alpha_{i_m}$ may issue multiple tool calls, and routing occurs only at segment boundaries.

### 3.2 Graph representation

A task graph is a labeled directed graph $G = (T, E, C)$ where $T = \{t_1, \ldots, t_n\}$ is a set of subtasks and $E = T \times T$ is the complete directed edge set (self-loops included). Each edge $(t_i, t_j) \in E$ carries a natural-language transition criterion $c_{i \to j} = C(t_i, t_j) \in \Sigma$ describing when control should move from $t_i$ to $t_j$ (e.g., "if verification fails, return to Execute"), where $\Sigma$ denotes natural-language strings. We use $t_i$ for the static node identity and $i_m$ for the runtime-selected node index at segment $m$ (full toy trace in a footnote).[1]

Although $E$ is complete by construction, the realized transition sequence is induced by router decisions and may be cyclic or acyclic; the goal is to make revisitation available and inspectable rather than to force cycles.

---

[1]Toy trace: let $T = \{t_1, t_2, t_3\}$ with $t_1 = $ SEARCH, $t_2 = $ PLAN, $t_3 = $ EXECUTE. A possible run selects indices $i_0 = 2$ (so $t_{i_0} = $ PLAN), $i_1 = 1$ (so $t_{i_1} = $ SEARCH), $i_2 = 1$ (self-loop on SEARCH), and $i_3 = 3$ (so $t_{i_3} = $ EXECUTE).

### 3.2.1 Agents and memories

We use three roles: planner $\pi$ (constructs $(T, C)$), executors $\{\alpha_i\}_{i=1}^n$ (each executes subtask $t_i$ with tool set $\mathcal{U}_i \subseteq \mathcal{U}$), and a unified analyzer+router $\rho$ (updates trajectory memory and selects the next node by criterion evaluation).

Let $\Delta k_m \in \mathbb{N}$ denote the number of tool calls issued by executor $\alpha_{i_m}$ within segment $m$. We maintain a global tool-call counter $k_m$ (at segment boundaries) with update

$$k_{m+1} = k_m + \Delta k_m, \qquad k_0 = 0. \tag{1}$$

Within each segment we enforce a local (per-segment) budget $\Delta k_m \leq C_l$. At the end of segment $m$ we maintain

$$S_m := (k_m, i_m, H_m, M_m), \tag{2}$$

where $k_m$ is the current global tool-call count, $i_m \in \{1, \ldots, n\}$ is the current node index, $H_m$ is the segment-level tool input/output summary (and raw trace) passed from the executor, and $M_m$ is a structured rolling memory maintained by $\rho$ and is passed to the next executor as soft guidance (the controller-level state variables and execution budgets are summarized in Appendix Secs. B and B.2).

### 3.2.2 Graph construction: Spec-Cyc vs. Gen-Cyc

We instantiate two construction regimes. Spec-Cyc constructs a task-specific graph per instance:

$$\pi_{\text{spec}} : (O, \mathcal{B}, \mathcal{D}) \mapsto (T, C), \tag{3}$$

where $\mathcal{B}$ is a benchmark description and $\mathcal{D} = \{d^{(r)}\}_{r=1}^n$ is the set of optional $n$ demonstration summaries of successful episode trajectories, and subtasks/criteria may reference instance-specific entities and failure modes. Gen-Cyc constructs a benchmark-generic graph,

$$\pi_{\text{gen}} : (\mathcal{B}, \mathcal{D}) \mapsto (T, C) \tag{4}$$

once per benchmark family and reuses it across instances to test transfer of cyclic revisitation rules.

### 3.2.3 Runtime execution and routing

Execution alternates between (i) running the current executor and (ii) routing to the next node. Given $(S_m, t_{i_m})$, the executor runs an inner tool-call loop and the router updates memory

$$M_{m+1} = \text{UPDATE}_\rho(M_m, t_{i_m}, H_m), \tag{5}$$

then selects the next node by criterion evaluation

$$i_{m+1} = \text{SELECT}_\rho(M_{m+1}, \{c_{i_m \to j}\}_{j=1}^n), \tag{6}$$

where self-loops ($i_{m+1} = i_m$) correspond to repeating the current subtask when criteria indicate "continue current work." Termination occurs if the environment signals success or the global tool-call budget is exhausted ($k_{m+1} \geq C_g$).

### 3.2.4 DepDAG baseline

To separate local retry from unrestricted cyclic revisitation, we evaluate DepDAG, a dependency-directed workflow in which the controller may either retry the current subtask, advance along the dependency graph, or terminate. In this paper, DepDAG denotes the dependency-directed baseline with local same-subtask retry unless explicitly marked as DepDAG-NoRetry. Let $\text{Succ}_{\text{DAG}}(i_m)$ be the set of DAG-allowed successors of the current subtask $i_m$, and let $\perp$ denote a terminal decision. The DepDAG candidate set is

$$\mathcal{A}_m^{\text{DAG}} = \text{Succ}_{\text{DAG}}(i_m) \cup \{i_m\} \cup \{\perp\}. \tag{7}$$

The analyzer/router selects

$$i_{m+1} = \text{SELECT}_\rho(M_{m+1}, \{c_{i_m \to j}\}_{j \in \mathcal{A}_m^{\text{DAG}}}),\tag{8}$$

where $i_{m+1} = i_m$ denotes retrying the same subtask and $i_{m+1} = \perp$ denotes termination. The self-loop is used when the current subtask is incomplete or locally recoverable, while inter-subtask movement remains constrained by the dependency graph. DepDAG therefore tests whether local repair within a dependency-directed workflow is sufficient, without granting the unrestricted cross-subtask revisitation available to complete cyclic graphs.

The runtime logs also make this structural distinction inspectable. For completed DepDAG runs, we record the dependency adjacency, allowed successors, typed transitions, and invalid-transition counts. Across the audited completed runs, all non-terminal transitions were either same-subtask retries or DAG-forward transitions (Appendix D.2).

### 3.3 Diagnostic workflow signatures and qualified workflow selection

The complete graph is a diagnostic instrument rather than a universal deployment prescription. Its purpose is to expose the reliability benefit and cost of unrestricted revisitation before a practitioner commits to a simpler or sparser controller. We therefore characterize the evaluated task families by a workflow signature: prerequisite density, state uncertainty, observability, and repair semantics. Table 1 summarizes the resulting diagnostic matrix.

This matrix is scoped to the benchmark case studies evaluated in this paper. It should not be read as evidence that a single benchmark fully represents a broad task regime, nor as a universal recommendation for all tasks with similar surface structure. Instead, it summarizes hypotheses suggested by the experiments: stable prerequisite-chain tasks appear to favor ReAct or DAG-like execution, partially observable interactive tasks appear to benefit from explicit recovery routes, and evidence-heavy open-world tasks require evidence-quality mechanisms beyond workflow control.

### 3.4 Computational overhead and scaling

A complete graph over $n$ subtasks stores $O(n^2)$ transition criteria. At runtime, however, a one-call router inspects only the $n$ outgoing criteria of the current node, so the number of router calls per segment remains one while routing-input length grows linearly in $n$ under bounded memory. The observed graphs are small, approximately 3–9 nodes with criteria of 7–33 words, and are therefore tractable at the evaluated scale. Larger workflows would motivate transition pruning or retrieval over candidate edges. Appendix H gives the detailed accounting, and Appendix Tables 21 and 23 report the observed graph sizes.

## 4 Experiments

### 4.1 Experimental Setup

This section describes the evaluation benchmarks, agent configurations, ablation studies, and reported metrics. Unless otherwise specified, every configuration in every ablation is run with three random seeds; we report mean and standard deviation over seeds. All agents are run using either `gpt-4o-mini` and/or `gpt-5-mini`.

#### 4.1.1 Benchmarks

We evaluate on three long-horizon tool-using benchmarks: TextCraft (depth 2/3/4 crafting tasks with increasing horizon) Prasad et al. (2024), ALFWorld (interactive household tasks emphasizing exploration and recovery) Shridhar et al. (2021), and Finance-Agent (open-world financial question answering with web research and evidence aggregation) Bigeard et al. (2025). A run is successful if the benchmark goal is achieved within the episode tool-call budget. For Finance-Agent, the publicly available validation set contains 50 cases; in our protocol, a fixed subset is used only for optional $n$-shot successful-trajectory summary construction,

Table 1: Diagnostic workflow signatures for the evaluated benchmark case studies. The workflow implication column summarizes hypotheses suggested by the benchmarks in this paper; it is not a universal architecture recommendation.

| Evaluated case study | Prerequisite density | Uncertainty / observability | Dominant failure mode observed | Workflow implication to test |
|---|---|---|---|---|
| TextCraft prerequisite-chain execution | High and mostly known | Low to moderate; state is inspectable | Skipping, misordering, or repeatedly attempting non-obtainable intermediates | ReAct, static DAGs, or DepDAG are strong candidates; complete cyclic graphs are useful mainly as a diagnostic stress test before pruning. |
| ALFWorld stateful recovery and exploration | Moderate; ordering depends on discovered state | High; partial observability and misleading actions | Mis-localization, failed manipulation, stale state, and need to revisit search/verification | Complete cyclic graphs or state machines with explicit recovery loops are plausible candidates; retain backtracking and verification edges. |
| Finance-Agent open-world evidence synthesis | Moderate and data-dependent | High; retrieved evidence is noisy/incomplete | Retrieval miss, source mismatch, weak grounding, unsupported synthesis | Workflow control should be paired with evidence-quality gates, retrieval auditing, and final-answer verification; cyclic routing alone is not expected to resolve evidence failures. |

and the remainder is used for held-out evaluation. Exact split details for all benchmarks, including exceptions such as TextCraft-4 and the benchmark-provided ALFWorld partitions, are given in Appendix B.1.

### 4.1.2 Methods Compared

We compare ReAct, DepDAG, Spec-Cyc, and Gen-Cyc. ReAct is a tool-using single-agent baseline that interleaves reasoning and actions in a closed loop. DepDAG is a dependency-directed workflow with local retry: a planner LLM generates an ordered subtask list and dependency DAG, and the analyzer is invoked between subtask segments to update memory and choose among the current subtask, DAG-allowed successor subtasks, and termination. This gives the dependency-directed controller local same-subtask retry while still disallowing arbitrary backtracking to previously visited or non-successor subtasks. Spec-Cyc and Gen-Cyc represent tasks as fully connected subtask graphs where each node is executed by a tool-using executor and a unified state-analysis-and-routing controller selects transitions by evaluating natural-language criteria. Spec-Cyc is generated per task instance, while Gen-Cyc is generated once per benchmark family and reused to test whether cyclic revisitation transfers across tasks. The earlier dependency-directed variant without local retry is retained in Appendix D.1 as DepDAG-NoRetry, an additional diagnostic variant.

### 4.1.3 Metrics

We report success rate (SR), successful-episode tool calls (TC), and efficiency STR = SR/TC. For cyclic runs, we additionally report subtask-visitation statistics on successful episodes: average total subtask visits (ATS), average unique subtasks visited (AUS), and average revisitation ASR = ATS − AUS. We also report unique transitions (UT) over full runs. For fault-injection experiments, we report the success-retention ratio and successful-episode tool-call overhead, $\mathrm{SRR} = \frac{\mathrm{SR_{fault}}}{\mathrm{SR_{nominal}}}, \Delta\mathrm{TC} = \mathrm{TC_{fault}} - \mathrm{TC_{nominal}}$. The current

Table 2: Overall comparison of ReAct, DepDAG, Spec-Cyc, and Gen-Cyc across benchmarks without $n$-shot summaries. SR is reported as mean percentage $\pm$ standard deviation across 3 seeds. TC is the average number of tool calls on successful episodes. STR is computed as SR/TC using SR in percentage points.

| | ReAct | | | DepDAG | | | Spec-Cyc | | | Gen-Cyc | | |
|---|---|---|---|---|---|---|---|---|---|---|---|---|
| Benchmark | SR (%) | TC | STR | SR (%) | TC | STR | SR (%) | TC | STR | SR (%) | TC | STR |
| TextCraft-2 | $94.3 \pm 0.2$ | $7.4 \pm 4.3$ | 12.7 | $88.0 \pm 0.9$ | $11.9 \pm 6.7$ | 7.4 | $85.2 \pm 0.8$ | $11.3 \pm 6.5$ | 7.5 | $93.9 \pm 1.3$ | $9.4 \pm 5.6$ | 10.0 |
| TextCraft-3 | $82.5 \pm 2.1$ | $16.0 \pm 8.8$ | 5.2 | $61.0 \pm 3.0$ | $26.3 \pm 11.0$ | 2.3 | $50.4 \pm 2.3$ | $21.6 \pm 11.8$ | 2.3 | $71.5 \pm 1.5$ | $22.1 \pm 11.9$ | 3.2 |
| TextCraft-4 | $45.5 \pm 14.8$ | $23.4 \pm 8.4$ | 1.9 | $36.4 \pm 7.4$ | $62.8 \pm 22.7$ | 0.6 | $15.2 \pm 8.6$ | $68.8 \pm 20.1$ | 0.2 | $36.4 \pm 0.0$ | $69.1 \pm 15.0$ | 0.5 |
| ALFWorld | $33.8 \pm 0.4$ | $12.9 \pm 7.5$ | 2.6 | $44.3 \pm 2.3$ | $34.7 \pm 21.0$ | 1.3 | $58.2 \pm 4.6$ | $30.2 \pm 18.5$ | 1.9 | $56.7 \pm 3.2$ | $28.7 \pm 19.9$ | 2.0 |
| Finance-Agent | $12.4 \pm 1.3$ | $3.0 \pm 0.9$ | 4.1 | $10.5 \pm 2.7$ | $3.1 \pm 1.2$ | 3.4 | $14.3 \pm 0.0$ | $5.3 \pm 2.2$ | 2.7 | $15.2 \pm 2.7$ | $6.0 \pm 2.5$ | 2.5 |

instrumentation records token counts and tool-call counts, not wall-clock latency; we therefore treat tokens and calls as compute proxies and identify latency instrumentation as a deployment requirement.

#### 4.1.4 Ablation Studies and Experimental Factors

We study six controlled factors. A1 is the main no-$n$-shot comparison among ReAct, DepDAG, Spec-Cyc, and Gen-Cyc. A2 varies planner, executor, and router model tiers to test orchestration sensitivity. A3 adds successful-trajectory summaries to test whether experience-derived routines improve planning and routing. A4 compares generalist full-tool executors with tool-restricted specialists on TextCraft and Finance-Agent. A5 injects random subtask redirections to stress-test recovery under disrupted control flow. A6 measures ALFWorld environment-level exploration through unique visited states. DepDAG-NoRetry is reported separately in Appendix D.1 to isolate the effect of same-subtask retry.

#### 4.1.5 Budgets, Hyperparameters, and Reproducibility

To comprehensively measure performance, we set global tool-call limits of 30, 50, and 100 for TextCraft depths 2, 3, and 4 respectively, 75 for ALFWorld, and 20 for Finance-Agent. The local subtask limit is fixed at 5 for TextCraft, 15 for ALFWorld, and 20 for Finance-Agent for cyclic and dependency-directed workflows. ReAct uses a unified limit equal to the benchmark global limit. Benchmark splits and evaluation budgets are summarized in Appendix Sec. B.2, while model roles, decoding controls, reasoning settings, and provider defaults are summarized in Appendix Table 8. Additional Finance-Agent model-coverage settings are reported with the corresponding sensitivity checks in Appendix B.5. Across experiments, we varied the method/controller and selected agent/model aliases, while leaving the reported decoding controls fixed unless explicitly stated otherwise. Some provider-side settings, including historical reasoning effort for several runs, were not recoverable from immutable run artifacts; we report such fields explicitly rather than inferring them. The supplementary artifact provided for review includes prompt templates, structured-output schemas, execution scripts, result-processing utilities, and anonymized logs needed to audit the reported aggregate results.

#### 4.1.6 Train/Test Split and $n$-Shot Construction

For TextCraft depths 2 and 3 and for Finance-Agent, we reserve a fixed subset of instances for constructing optional $n$-shot successful-trajectory summaries and evaluate on the remaining held-out instances. For TextCraft-4, only 11 instances are available, so all 11 are used for evaluation and no separate summary-construction split is formed. For ALFWorld, we follow the benchmark's provided partitions: benchmark training games are used only for optional summary construction.

Concretely, when $n$-shot summaries are enabled, we run Spec-Cyc on the designated summary-construction subset, summarize successful trajectories, and then freeze that summary set for the rest of the benchmark. All reported results are on held-out evaluation instances; the only difference in the $n$-shot setting is that the planner receives the fixed summary set as additional context. Exact benchmark counts are listed in Appendix B.2.

## 4.2 Experimental Results

We organize results around the uniform head-to-head without $n$-shot summaries (Table 2), graph/role sensitivity with $n$-shot summaries (Fig. 2), robustness under control-flow perturbation (Table 3), and success-conditioned revisitation (Fig. 3). Remaining sweeps, DepDAG-NoRetry diagnostics, DepDAG transition audits, routing-complexity statistics, uncertainty summaries, graph-stability details, and qualitative cases are summarized in the appendix.

**A1: Overall comparison without $n$-shot summaries.** Table 2 gives the cleanest apples-to-apples comparison of single-agent, dependency-directed local-retry, and complete cyclic control: generalist executors, no successful-trajectory summaries, and `gpt-4o-mini` across all non-ReAct roles. The benchmarks separate clearly by structure. TextCraft is largely a prerequisite-chain domain, so ReAct remains strongest overall in both success and efficiency. DepDAG is competitive among explicit-workflow controllers on TextCraft, especially compared with the cyclic variants at higher depth, but its lower STR indicates that explicit decomposition and local retry still add coordination overhead in this relatively inspectable domain.

ALFWorld shows a different pattern. DepDAG improves over ReAct in success rate, indicating that explicit subtask structure and local retry are useful under partial observability. However, both complete cyclic variants outperform DepDAG in SR, consistent with the need for broader backtracking, search, and recovery routes beyond same-subtask retry. Finance-Agent is different again: absolute success remains low across all methods, and the small differences among controllers should be read as contrastive rather than conclusive. In this benchmark, workflow structure alone appears insufficient without stronger retrieval, grounding, and evidence-synthesis mechanisms.

**A2/A3: Ablations across graph regimes and role allocations ($n$-shot).** Figure 2 compares success-rate sensitivity to graph regime and planner–executor–router allocation using the three representative allocations available for both graph regimes on all five benchmarks. Error bars show standard deviation across three seeds. Role allocation materially affects cyclic performance, with stronger execution and routing often producing the largest gains at longer horizons. Gen-Cyc is frequently competitive with Spec-Cyc and is substantially stronger in several TextCraft settings, although Spec-Cyc remains preferable in some configurations. The complete sweep, including additional mixed allocations and TC, STR, ATS, AUS, and ASR, appears in Appendix Table 15.

**A4: Tool exposure.** We additionally vary executor tool exposure on TextCraft and Finance-Agent, comparing generalist (full-tool) vs. tool-restricted specialized executors. We omit ALFWorld from the specialization sweep because, unlike TextCraft (naturally separable `get`/`craft`-style interactions) and Finance-Agent (pre-packaged multiple tools), ALFWorld exposes a single, monolithic action interface; any "tool" partition would be an arbitrary design choice with many plausible alternatives. As a result, specialist performance on ALFWorld would likely be sensitive to the chosen partitioning scheme, so we exclude it to avoid biasing the comparison. Tool restriction is consistently harmful here: specialized executors substantially reduce SR/STR on TextCraft and Finance-Agent, with the largest degradation under Spec-Cyc. Gen-Cyc is more robust under tool restriction and remains the preferred cyclic regime when tools are limited (Appendix Tables 16, 17, and 18).

**Failure-mode structure.** Trace-derived Finance-Agent diagnostics distinguish retrieval/source misses, grounding mismatches, budget exhaustion, routing loops, and unresolved failures (Appendix B.6 and Appendix I). Because these deterministic labels are diagnostic rather than human-adjudicated correctness judgments, we use them to characterize the evidence pipeline rather than rank controllers. They reinforce that the small Finance-Agent controller differences should be interpreted cautiously.

## 4.3 Robustness and recovery efficiency

The fault-injection study is treated as a robustness test for disrupted control flow. At segment boundaries, the nominal router is randomly overridden with a 50% probability and redirected to a different subtask. This perturbation is intentionally harsh: it does not model a realistic deployment policy, but instead tests whether a

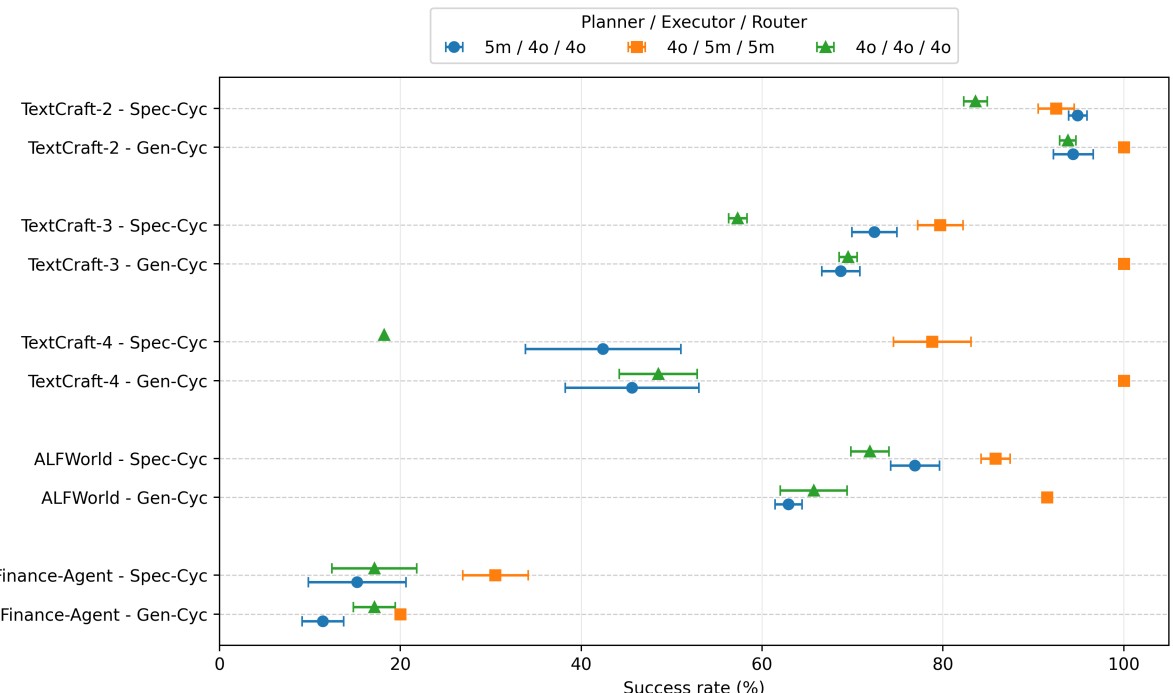

Figure 2: Success-rate sensitivity of Spec-Cyc and Gen-Cyc with $n$-shot summaries and generalist executors. Points show means and horizontal error bars show standard deviations across three seeds. Configurations are ordered as planner/executor/router; `5m` denotes `gpt-5-mini` and `4o` denotes `gpt-4o-mini`. The figure shows the three allocations available for both graph regimes on every benchmark; the complete role-allocation sweep and all cost and visitation metrics appear in Appendix Table 15.

controller can retain episode-level success when forced off its nominal route. A dependency-directed controller such as DepDAG permits local retry and forward recovery, but it does not permit arbitrary cross-subtask backtracking unless those exceptions are explicitly encoded. In contrast, the complete cyclic graph keeps every subtask reachable, so robustness can be measured as success retention under randomized redirection.

Table 3 summarizes the generalist-executor setting. The strongest reliability signal is Gen-Cyc on TextCraft: despite perturbation, it retains 96.1% of nominal success on TextCraft-2 and 94.1% on TextCraft-3, with only 1.2 and 3.4 additional successful-episode tool calls, respectively. Spec-Cyc is less stable under the same perturbation, dropping to 59.1% retention on TextCraft-2 and 34.0% retention on TextCraft-3. On ALFWorld, both cyclic variants retain substantial success under perturbation, consistent with the benchmark's need for search and recovery. These results quantify the core workflow-control trade-off: flexible routing is not free, but in regimes where the generic transition rules are stable, a small call overhead can buy high fault tolerance.

This is still an episode-level robustness measure rather than an event-level repair-latency measure. The present traces do not separately log "fault occurrence → corrected state" intervals, so we cannot claim a precise number of cycles or seconds per individual recovery event. The available evidence supports a deployment-relevant conclusion: under harsh randomized routing disruption, some cyclic controllers, especially Gen-Cyc in TextCraft, retain most of their nominal success with modest successful-episode call overhead, whereas a forward-only controller would require externally specified exception handling or restart logic.

**Success-conditioned revisitation.** Successful cyclic trajectories generally contain both first visits and revisitation, although conditioning on success cannot distinguish corrective revisits from wasteful detours. Appendix Fig. 3 reports the full visitation decomposition, and the qualitative traces illustrate both productive recovery and unproductive looping.

Table 3: Episode-level recovery efficiency under control-flow fault injection for generalist executors (all components `gpt-4o-mini`). SRR is perturbed success divided by nominal success; $\Delta$TC is the change in successful-episode tool calls.

| Benchmark | Graph | SRR | $\Delta$TC | Interpretation |
|---|---|---|---|---|
| TextCraft-2 | Spec-Cyc | 0.591 | +7.5 | Task-specific criteria are brittle when forced off-route. |
| TextCraft-2 | Gen-Cyc | 0.961 | +1.2 | Generic rules preserve most success with small recovery overhead. |
| TextCraft-3 | Spec-Cyc | 0.340 | +12.3 | Perturbations amplify loop and misrouting cost. |
| TextCraft-3 | Gen-Cyc | 0.941 | +3.4 | Strong retention despite deeper prerequisite chains. |
| TextCraft-4 | Spec-Cyc | 0.000 | D/A | Low nominal success leaves little recoverable margin. |
| TextCraft-4 | Gen-Cyc | 0.874 | −6.2 | Perturbed successes are fewer but not more call-intensive. |
| ALFWorld | Spec-Cyc | 0.865 | −1.5 | Recovery routes remain useful under partial observability. |
| ALFWorld | Gen-Cyc | 0.769 | −1.6 | Success drops more than Spec-Cyc, but retained wins remain efficient. |

### 4.4 Finance-Agent as an evidence-synthesis pipeline

Finance-Agent remains a low-success setting across controllers: ReAct reaches 12.4%, DepDAG 10.5%, Spec-Cyc 14.3%, and Gen-Cyc 15.2% in the main no-$n$-shot comparison (Table 2). Given the small held-out set and close absolute results, these differences are contrastive rather than conclusive. Additional Gemini 2.5 Pro and DeepSeek-v3.1-Terminus checks likewise remain low-success and do not establish a cyclic-controller advantage (Appendix B.5). We therefore treat Finance-Agent as an evidence-pipeline stress test: revisitation is useful only when it improves retrieval, preserves relevant evidence, or resolves grounding failures; otherwise, it adds routing and token cost without addressing the underlying information gap.

## 5 Discussion

Complete cyclic subtask graphs are best understood here as a deliberately maximally flexible multi-agent regime rather than as a claim that fully connected control is always optimal. Because every executable subtask can transition to every other subtask, the controller is never forced into a predefined trajectory: recovery, revisitation, and exploratory detours are always structurally available. That design choice is exactly what makes the method informative. It lets us observe when flexibility translates into useful recovery, when it turns into coordination overhead or router hallucination, and when the main bottleneck lies outside the workflow controller altogether.

**Benchmark structure determines whether flexibility is an asset or a tax.** The three benchmarks expose three different regimes. In ALFWorld, partial observability and interaction-heavy search make backtracking genuinely useful, so cyclic control shows its clearest advantage. In TextCraft, once the correct prerequisite chain is identified, the problem is closer to disciplined forward execution, so the same routing flexibility can become extra coordination cost. Finance-Agent exposes a third regime: open-world retrieval and evidence aggregation dominate, so even a flexible multi-agent controller yields only modest gains because the harder problem is grounding and synthesis, not just workflow navigation.

**Cost matters, and model strength changes what can be inferred.** The central controlled setting uses weaker-tier models to expose orchestration effects that may be hidden when a single strong agent already solves most cases. The point is not simply to maximize benchmark scores, but to measure whether multi-agent flexibility can recover mistakes that a simpler controller makes. This is also the regime where the tradeoff is visible: cyclic control can help, but it often requires substantially more inference. However, when absolute performance remains low, as in Finance-Agent, weak-model results alone cannot establish the dominant bottleneck. For this reason, we report Finance-Agent as an evidence-pipeline stress test and use additional model-coverage checks as sensitivity analyses rather than definitive leaderboard comparisons.

**Why Gen-Cyc can sometimes outperform Spec-Cyc.** Benchmark-generic graphs can act as a regularizing bias because their subtasks and routing criteria are typically simpler, reusable, and less entangled with instance-specific wording. Spec-Cyc can be more expressive, but that same expressivity can also produce

brittle high-precision transitions, especially when the router over-trusts subtask intent relative to the actually observed state. Our results therefore do not suggest that Gen-Cyc is uniformly superior; rather, they indicate that the simpler reusable regime is often competitive and, in several long-horizon or perturbed settings, more stable and recoverable than task-specific graphs.

**From diagnostic completeness to sparse deployment.** Complete connectivity is an upper-flexibility condition, not a deployment prescription. The transition analyses identify never-won and recurrent loop-inducing edges that could be pruned or downweighted (Appendix Tables 19 and 20). Sparse cyclic graphs and hand-designed recovery state machines are therefore natural distillation targets, but evaluating them further requires benchmark-specific engineering, supervision, or post-hoc transition-utility estimates.

## 6 Limitations and Reproducibility Notes

The experiments characterize workflow-control behavior under the prompts, tools, model tiers, decoding settings, recovered reasoning-setting fields, and budgets reported in the appendix, without implying that complete cyclic graphs are universally optimal. The current logs support token and tool-call cost accounting, but they do not contain wall-clock latency traces; therefore, latency is discussed as a deployment instrumentation requirement rather than reported as a measured metric. Similarly, the fault-injection analysis uses episode-level recovery proxies rather than per-failure cycle-to-recovery traces. We therefore describe those results as success-retention under disrupted control flow, not as fine-grained recovery-latency measurements. The benchmark coverage is limited. Each evaluated benchmark is a case study for a workflow signature, not a definitive representative of all prerequisite-chain, partially observable, or evidence-synthesis tasks. Finance-Agent in particular should not be interpreted as proving that retrieval and grounding dominate in all financial workflows; rather, the low absolute success and model-coverage checks indicate that workflow control alone is insufficient under the current benchmark tools and budget. An anonymized repository containing the prompts, schemas, scripts, and utilities is provided for review and will be released publicly upon acceptance.

## 7 Broader Impact Statement

This work studies workflow-control structures for tool-using LLM agents. The main positive impact is improved diagnosis of when flexible revisitation and recovery are worth their cost in long-horizon automation. The same mechanisms could be misapplied to high-stakes domains if users treat successful task completion as a substitute for evidence quality, domain validation, or human oversight. The Finance-Agent results argue against such over-reliance: flexible revisitation is not a substitute for evidence verification, domain expertise, or human review in financial, legal, medical, or safety-critical workflows. In such settings, workflow controllers should be paired with source validation, uncertainty reporting, audit trails, and appropriate human oversight.

## 8 Conclusions

We studied complete cyclic subtask graphs as a maximally flexible workflow-control design for automated large-language-model workflows. The value of this design is not that full connectivity is universally best, but that it makes the flexibility–cost–reliability tradeoff directly measurable at the level of executable subtasks. The qualified workflow-signature matrix summarizes hypotheses suggested by the evaluated case studies: cyclic control is most useful when recovery and exploration are central, can add overhead in stable prerequisite-chain tasks, and cannot by itself solve evidence-quality bottlenecks in open-world financial research. DepDAG shows that local retry can close part of the gap between dependency-directed and cyclic workflows, while ALFWorld still benefits from broader revisitation beyond same-subtask retry. The robustness results further show that generic cyclic transition rules can retain high success under harsh routing perturbations with modest successful-episode call overhead, while task-specific high-precision routes can be brittle. Overall, complete cyclic subtask graphs provide a diagnostic lens for deciding when multi-agent flexibility is worth its price and when practitioners should instead deploy a simpler directed acyclic graph, local-retry controller, state machine, or sparsified cyclic controller.

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

# A Protocol, Key Tables, and Case Studies

This appendix retains the material most relevant for auditing the main claims in the paper, including protocol details, model settings, token accounting, ablation tables, graph-stability summaries, transition audits, routing-complexity statistics, and qualitative traces. Prompt templates, structured-output schemas, execution scripts, result-processing utilities, and anonymized logs are included in the supplementary artifact.

# B Benchmark and protocol details

## B.1 Benchmark Details

We evaluate on three long-horizon tool-using benchmarks that differ in environment dynamics, observability, and failure modes.

### B.1.1 TextCraft

**TextCraft.** TextCraft is a text-based crafting environment that requires multi-step composition of actions and intermediate prerequisites. We use three difficulty tiers (TextCraft-2/3/4), where higher indices correspond to deeper dependency chains and longer horizons. We treat each instance as successful if the target crafting goal is achieved within the allowed tool-call budget.

The tool set supports item acquisition, crafting operations, and inventory management.

**textcraft_inventory**
Check the current contents of the agent's inventory.

- **Parameters:** None
- **Returns:** A listing of all items currently in the inventory with their quantities

**textcraft_get_item**
Acquire items from the environment. Not all items can be obtained directly—some must be crafted.

- **Parameters:**
  - `reasoning` (List[str]): Step-by-step reasoning assessing which item to get
  - `item_name` (Literal): The name of the item to acquire, constrained to obtainable items
  - `num_total_items_needed` (int, $\geq 1$): The quantity of items to acquire
- **Returns:** The result of the acquisition attempt

**textcraft_craft**
Craft items using materials in the inventory according to available recipes.

- **Parameters:**
  - `reasoning` (List[str]): Step-by-step reasoning assessing which item to craft
  - `crafting_command` (Literal): The crafting command in the format "craft *N item* using *M ingredient1*, ...", constrained to valid recipes
- **Returns:** The result of the crafting action, including success or failure with explanation

**textcraft_select_command**
A unified command interface combining all TextCraft operations.

- **Parameters:**
  - `reasoning` (List[str]): Step-by-step reasoning assessing which command to execute
  - `command` (str): One of:
    * `get <number> <item>` – Acquire items from the environment
    * `craft <crafting_command>` – Craft items using inventory materials
    * `inventory` – Check inventory contents
- **Returns:** The observation after executing the command

### B.1.2 ALFWorld

**ALFWorld.** ALFWorld is an interactive household environment where agents must navigate, manipulate objects, and complete multi-step tasks. Compared to TextCraft, ALFWorld places greater emphasis on exploration, partial observability, and recovery after incorrect actions. We treat each episode as successful if the specified goal condition is satisfied within the tool-call budget.

The tool set is dynamically constrained based on the current game state, only valid commands for the current context are exposed to the agent.

**alfworld_select_command**
Select and execute a command from the list of currently admissible commands.

- **Parameters:**
  - `reasoning` (List[str]): Step-by-step reasoning assessing which command to execute
  - `command` (Literal): The command to execute, constrained to the set of admissible commands
- **Returns:** The next observation of the environment after executing the command

**alfworld_manage_inventory**
Manage the agent's inventory by checking contents, taking items from receptacles, or placing items.

- **Parameters:**
  - `reasoning` (List[str]): Step-by-step reasoning assessing which inventory command to execute
  - `command` (Literal): One of:
    * `inventory` – Check current inventory contents
    * `take <object> from <receptacle>` – Take an item from a receptacle
    * `put <object> in/on <receptacle>` – Place an item in or on a receptacle
    * `examine <inventory_item>` – Examine an item in inventory
- **Returns:** The observation after the inventory operation

**alfworld_explore**
Navigate and explore the environment to discover objects and locations.

- **Parameters:**
  - `reasoning` (List[str]): Step-by-step reasoning about which navigational command to execute
  - `command` (Literal): One of:
    * `look` – Look around the current location
    * `go to <receptacle>` – Move to a specific furniture or receptacle
    * `examine <receptacle>` – Get detailed information about a receptacle
    * `open <receptacle>` – Open a closed receptacle
    * `close <receptacle>` – Close an open receptacle
- **Returns:** The observation after the exploration action

**alfworld_manipulate_object**
Perform physical manipulations on objects using tools or appliances.

- **Parameters:**
  - `reasoning` (List[str]): Step-by-step reasoning assessing which manipulation to perform
  - `command` (Literal): One of:
    * `use <object>` – Use an item in inventory
    * `heat <object> with <appliance>` – Heat an item using an appliance (e.g., microwave)
    * `cool <object> with <appliance>` – Cool an item using an appliance (e.g., fridge)
    * `clean <object> with <receptacle>` – Clean an item using a receptacle (e.g., sink)
    * `slice <object> with <tool>` – Cut an item using a tool (e.g., knife)
- **Returns:** The observation after the manipulation

### B.1.3 Finance-Agent

**Finance-Agent.** Finance-Agent consists of information-seeking tasks requiring web research and evidence aggregation to produce a final, grounded financial answer. The environment is open-world and tool interactions can return large, noisy observations. An episode is successful if the produced final answer matches the benchmark's correctness criteria under the allowed tool-call budget. We evaluate only on the publicly available validation split, which contains 50 cases, and report success under the same tool-call budget and evaluation criteria as in the benchmark.

The tool set enables systematic gathering and processing of financial data.

**finance_agent_edgar_search**
    Search the SEC EDGAR database for regulatory filings.

- **Parameters:**
    - `query` (str): Keyword or phrase to search (e.g., "substantial doubt" OR "material weakness")
    - `form_types` (List[str], optional): Filter by form types (e.g., ["10-K", "10-Q", "8-K"])
    - `ciks` (List[str], optional): Filter by company CIK numbers
    - `start_date` (str, optional): Start date in yyyy-mm-dd format
    - `end_date` (str, optional): End date in yyyy-mm-dd format
    - `page` (int, default=1): Page number for pagination
    - `top_n_results` (int, default=10): Maximum number of results to return
- **Returns:** Filing metadata including company name, ticker, CIK, form type, filing date, description, and document URL

**finance_agent_google_web_search**
    Search the web for financial information using Google Search.

- **Parameters:**
    - `search_query` (str): The query to search for on the web
- **Returns:** Summarized search results with relevant information extracted

**finance_agent_parse_html_page**
    Fetch and parse an HTML page, storing the extracted text content for later retrieval.

- **Parameters:**
    - `url` (str): The URL of the HTML page to parse
    - `key` (str): The key under which to store the parsed content for later retrieval
- **Returns:** Confirmation of success with content length and current data storage keys

**finance_agent_retrieve_information**
    Retrieve and summarize previously stored documents using placeholder substitution.

- **Parameters:**
    - `prompt` (str): A prompt containing `{{key_name}}` placeholders that will be replaced with stored content (e.g., "Summarize this filing: {{company_10k}}")
- **Returns:** The prompt with placeholders replaced by summarized document content

**finance_agent_final_answer**
    Submit the final answer to the financial question.

- **Parameters:**
    - `answer` (str): The final answer to the financial question
    - `sources` (List[SourceInfo], optional): List of sources with URLs and descriptive names
- **Returns:** Confirmation that the answer has been submitted

## B.2 Experimental Protocol Overview

This appendix documents the train/test splits used for each benchmark. All splits are deterministically shuffled using a fixed random seed of 42 to ensure reproducibility.

### B.2.1 Summary

Table 4: Overview of benchmark dataset sizes and train/test splits.

| Benchmark | Total Seeds | Train Set | Test Set |
|---|---|---|---|
| TextCraft (Depth 2) | 291 | 88 (30%) | 203 (70%) |
| TextCraft (Depth 3) | 117 | 35 (30%) | 82 (70%) |
| TextCraft (Depth 4) | 11 | 0 (0%) | 11 (100%) |
| ALFWorld | 3,827 | 3,553 (train) | 134 (unseen) / 140 (seen) |
| Finance-Agent | 50 | 15 (30%) | 35 (70%) |

### B.2.2 TextCraft

TextCraft tasks are stratified by recipe depth, representing the number of intermediate crafting steps required to produce the target item. Each depth level constitutes an independent dataset with its own train/test split.

Table 5: TextCraft dataset splits by recipe depth.

| Depth | Total | Train | Test | Test Partitions |
|---|---|---|---|---|
| 2 | 291 | 88 | 203 | 5 partitions (45, 45, 45, 45, 23 seeds) |
| 3 | 117 | 35 | 82 | 2 partitions (45, 37 seeds) |
| 4 | 11 | 0 | 11 | 1 partition (11 seeds) |

Seeds are shuffled deterministically before splitting. The train set comprises the first 30% of shuffled seeds, with the remainder allocated to the test set. Depth 4 contains only 11 seeds due to the limited number of recipes requiring four crafting steps; all are used for evaluation.

### B.2.3 ALFWorld

ALFWorld provides pre-defined dataset splits through its environment configuration, derived from the ALFRED dataset Shridhar et al. (2020). The benchmark uses TextWorld-based game files generated from ALFRED trajectories.

Table 6: ALFWorld dataset configurations.

| Dataset | Games | Description |
|---|---|---|
| train | 3,553 | Training set |
| eval_in_distribution | 140 | In-distribution evaluation |
| eval_out_of_distribution | 134 | Out-of-distribution evaluation |

The primary evaluation uses the `eval_out_of_distribution` dataset (also known as `valid_unseen`) containing 134 test cases with novel scene and object configurations not seen during training.

The `eval_in_distribution` dataset (`valid_seen`) contains 140 games with scene configurations similar to the training set, used for ablation studies.

### B.2.4   Finance-Agent

Finance-Agent consists of the 50 publicly available financial question-answering tasks requiring retrieval and analysis of SEC filings and financial documents.

Table 7: Finance-Agent dataset split.

| Split | Seeds | Percentage |
|-------|-------|------------|
| Train | 15 | 30% |
| Test | 35 | 70% |
| **Total** | 50 | 100% |

### B.2.5   Reproducibility

All random shuffling operations use Python's `random` module with a fixed seed of 42:

```
random.seed(42)
seeds = list(range(total_seeds))
random.shuffle(seeds)
train_seeds = seeds[:train_size]
test_seeds = seeds[train_size:]
```

This ensures identical train/test splits across all experimental runs.

### B.3   Model, decoding, and reasoning settings

Table 8 reports the model, decoding, reasoning, and budget settings used across the experiments. Across runs, we varied the controller/method and selected agent/model aliases, but did not otherwise tune decoding hyperparameters between experiments. When top-$p$, top-$k$, or related sampling controls are listed as provider default, this means they were not explicitly set in our run configuration.

Table 8: Model, decoding, reasoning, and budget settings used in the reported experiments. Provider default means the field was not explicitly set in the run configuration.

| Setting | Value |
|---------|-------|
| Main OpenAI task/executor models | `gpt-4o-mini` and `gpt-5-mini`, selected by the corresponding agent/model alias. |
| Main OpenAI graph-generation model | `gpt-4o-mini` or `gpt-5-mini`, depending on the graph-generation alias used for the configuration. |
| Main OpenAI summarizer/router model | `gpt-4o-mini` or `gpt-5-mini`, depending on the summarizer/router alias used for the configuration. |
| Task/executor temperature | 0.0. |
| Summarizer/router temperature | 0.0. |
| Graph-generator temperature | 1.0. |
| `gpt-5-mini` reasoning effort | Low. |
| `gpt-4o-mini` reasoning effort | Not applicable / not set. |
| Top-$p$, top-$k$, and related sampling controls | Not explicitly set; provider defaults used throughout. |
| Task/executor output cap | `max_tokens=2048`. |
| Summarizer/router output cap | `max_tokens=4096`. |
| Graph-generator output cap | `max_tokens=16384` for subtask JSON and flowchart JSON generation. |
| TextCraft global tool-call budget | 30, 50, and 100 for TextCraft depths 2, 3, and 4, respectively. |
| ALFWorld global tool-call budget | 75. |
| Finance-Agent global tool-call budget | 20. |
| TextCraft local/subtask cap | 5. |
| ALFWorld local/subtask cap | 15. |
| Finance-Agent local/subtask cap | 20 from the runner; the execution loop uses min(`task_agent.max_tool_calls`, 20). |
| ReAct tool-call budget | ReAct uses the benchmark global tool-call budget as a unified action budget. |
| Finance select-command agent max tool calls | 20. |
| Finance specialized-agent max tool calls | Mostly the default `max_tool_calls` = 5, unless explicitly overridden by the selected agent configuration. |
| LLM judge agent | `agent.gpt_5_low`: gpt-5, `temperature=None`, `reasoning_effort=low`. |
| LLM judge top-$p$ / output cap | Not explicitly set; provider defaults used. |

### B.4 Uncertainty analysis for small held-out sets

For small held-out sets, especially Finance-Agent with $n = 35$ test cases per seed, small absolute differences in success rate can be within sampling uncertainty. Where raw success counts are available, we report Wilson 95% confidence intervals for binomial success:

$$\hat{p}_{\mathrm{W}} = \frac{\hat{p} + z^2/(2n)}{1 + z^2/n}, \qquad h_{\mathrm{W}} = \frac{z}{1 + z^2/n}\sqrt{\frac{\hat{p}(1 - \hat{p})}{n} + \frac{z^2}{4n^2}}, \tag{9}$$

where $\hat{p} = x/n$, $x$ is the number of successful episodes, $n$ is the number of evaluated episodes, $z = 1.96$, and the interval is $[\hat{p}_{\mathrm{W}} - h_{\mathrm{W}}, \hat{p}_{\mathrm{W}} + h_{\mathrm{W}}]$. We use these intervals to avoid over-interpreting close Finance-Agent differences, and we treat broader model-coverage runs as sensitivity checks unless paired case-level tests support a stronger claim.

### B.5 Finance-Agent model-coverage checks

Table 9 reports Finance-Agent model-coverage checks using Gemini 2.5 Pro and DeepSeek-v3.1-Terminus accessed through OpenRouter. We distinguish the model family/source from the API access path because the purpose of this check is to test whether the Finance-Agent conclusion is merely an artifact of the OpenAI mini-model setting. These runs use the same 35-case held-out Finance-Agent split and the same global tool-call budget as the main Finance-Agent experiments. We report aggregate wins over the completed held-out evaluations and Wilson confidence intervals. These checks are intended as sensitivity analyses for the evidence-pipeline interpretation, not as leaderboard claims.

Table 9: Finance-Agent model-coverage checks on the held-out split. Wilson intervals are computed over aggregate wins across completed held-out evaluations. TC(success) is averaged over successful episodes; TC(all) is averaged over all episodes.

| Controller | Planner/router | Executor/ReAct model family | Access path | Wins | SR (Wilson 95% CI) | TC(success) | TC(all) |
|---|---|---|---|---|---|---|---|
| ReAct | – | Gemini 2.5 Pro | OpenRouter | 16/70 | 22.9% [14.6, 34.0] | $6.75 \pm 4.48$ | $8.20 \pm 5.70$ |
| Gen-Cyc | gpt-4o-mini / gpt-4o-mini | Gemini 2.5 Pro | OpenRouter | 12/70 | 17.1% [10.1, 27.6] | $9.50 \pm 4.58$ | $15.26 \pm 5.24$ |
| ReAct | – | DeepSeek-v3.1-Terminus | OpenRouter | 1/70 | 1.4% [0.2, 7.7] | $5.00 \pm 0.00$ | $12.50 \pm 7.55$ |
| Gen-Cyc | gpt-4o-mini / gpt-4o-mini | DeepSeek-v3.1-Terminus | OpenRouter | 2/70 | 2.9% [0.8, 9.8] | $11.50 \pm 0.71$ | $18.87 \pm 3.01$ |

The Gemini checks improve over the main `gpt-4o-mini` ReAct setting but do not saturate Finance-Agent, and the Gen-Cyc controller with a Gemini executor does not outperform Gemini ReAct in aggregate success. DeepSeek performs poorly in both controllers; because its traces show substantial tool/interface failures, we interpret those runs cautiously as additional provider-coverage checks rather than as evidence about the upper bound of model capability. The paired overlap analysis, matched by question text, likewise does not support a strong controller-ranking claim: Gemini ReAct and Gemini Gen-Cyc share seven wins, with three ReAct-only wins and one Gen-Cyc-only win after collapsing repeated attempts by question.

### B.6 Finance-Agent tool-use and failure diagnostics

Table 10 summarizes tool-use and evidence-pipeline diagnostics for the Finance-Agent model-coverage checks. The diagnostics are extracted from stored run artifacts and should be interpreted as trace-level indicators rather than human adjudications of financial correctness. The main pattern is that successful completion remains limited even when stronger or different-provider executors are used. Gemini ReAct submits final answers more often than the other settings, while Gen-Cyc with Gemini uses more tools and sources but still remains low-success. DeepSeek runs show high budget-exhaustion rates and low final-answer submission rates, so we interpret them cautiously as additional provider-coverage checks rather than as a strong model-capability comparison.

Table 11 reports heuristic trace labels extracted from the same stored runs. These labels are not human-validated adjudications, but they help separate evidence-acquisition, grounding, routing, and budget-related patterns. Gemini failures include retrieval/source misses, grounding mismatches, and routing loops, while

Table 10: Finance-Agent tool-use diagnostics for model-coverage checks. Rates are fractions of held-out episodes.

| Controller / model | Wins | Final-answer rate | Source-submission rate | Budget-exhaustion rate | Mean tool calls |
|---|---|---|---|---|---|
| ReAct / Gemini 2.5 Pro | 16/70 | 0.657 | 0.414 | 0.114 | 8.20 |
| Gen-Cyc / Gemini 2.5 Pro | 12/70 | 0.586 | 0.471 | 0.443 | 15.26 |
| ReAct / DeepSeek-v3.1-Terminus | 1/70 | 0.029 | 0.029 | 0.500 | 12.50 |
| Gen-Cyc / DeepSeek-v3.1-Terminus | 2/70 | 0.171 | 0.057 | 0.829 | 18.87 |

both DeepSeek settings show very low final-answer submission and high budget-exhaustion behavior. We therefore use these runs as coverage checks and avoid drawing a strong ranking conclusion from them.

Table 11: Heuristic Finance-Agent trace diagnostics for model-coverage checks. Labels are trace-derived and not human-validated. "Other / unresolved" covers failed episodes whose stored traces did not support one of the more specific labels.

| Setting | Failed episodes | Budget/no final | Retrieval miss | Grounding mismatch | Routing loop | Other / unresolved |
|---|---|---|---|---|---|---|
| ReAct / Gemini 2.5 Pro | 54 | 2 | 13 | 7 | 5 | 27 |
| Gen-Cyc / Gemini 2.5 Pro | 58 | 0 | 0 | 0 | 2 | 56 |
| ReAct / DeepSeek-v3.1-Terminus | 69 | 0 | 31 | 0 | 0 | 38 |
| Gen-Cyc / DeepSeek-v3.1-Terminus | 68 | 0 | 0 | 0 | 0 | 68 |

## C  Shared-win token-cost comparison

Table 12: Relative token usage of cyclic workflows against ReAct. Each comparison block reports average total tokens per test case on the shared-win intersection (primary comparison) and on all won cases (secondary context). 'Excess' and 'Rel. diff.' are computed with respect to the shared-win row unless separately populated for the all-win row.

| Benchmark | Compared pair | Scope | ReAct avg total | Cyclic avg total | Excess (C−R) | Rel. diff. (%) |
|---|---|---|---|---|---|---|
| TextCraft-2 | ReAct (4o-mini) vs Spec-Cyc, generalist, P/E/R=4o/4o/4o | Shared wins | 81084.08 | 93208.69 | 12124.61 | 14.95 |
| | | All wins | 83360.6 | 93835.68 | 10475.08 | 12.57 |
| TextCraft-2 | ReAct (4o-mini) vs Gen-Cyc, generalist, P/E/R=4o/4o/4o | Shared wins | 81820.76 | 77694.46 | -4126.3 | -5.04 |
| | | All wins | 83360.6 | 78741.56 | -4619.04 | -5.54 |
| TextCraft-2 | ReAct (5-mini) vs Spec-Cyc, generalist, P/E/R=4o/5/5 | Shared wins | 22341.6 | 97919.85 | 75578.25 | 338.28 |
| | | All wins | 22490.85 | 97919.85 | 75429.0 | 335.38 |
| TextCraft-2 | ReAct (5-mini) vs Gen-Cyc, generalist, P/E/R=4o/5/5 | Shared wins | 22490.85 | 108678.92 | 86188.07 | 383.21 |
| | | All wins | 22490.85 | 108678.92 | 86188.07 | 383.21 |
| TextCraft-3 | ReAct (4o-mini) vs Spec-Cyc, generalist, P/E/R=4o/4o/4o | Shared wins | 109047.05 | 245365.14 | 136318.09 | 125.01 |
| | | All wins | 197838.72 | 252334.26 | 54495.54 | 27.55 |
| TextCraft-3 | ReAct (4o-mini) vs Gen-Cyc, generalist, P/E/R=4o/4o/4o | Shared wins | 172396.49 | 210395.97 | 37999.48 | 22.04 |
| | | All wins | 197838.72 | 209554.94 | 11716.22 | 5.92 |
| TextCraft-3 | ReAct (5-mini) vs Spec-Cyc, generalist, P/E/R=4o/5/5 | Shared wins | 57238.43 | 226266.16 | 169027.73 | 295.3 |
| | | All wins | 61405.6 | 226266.16 | 164860.56 | 268.48 |
| TextCraft-3 | ReAct (5-mini) vs Gen-Cyc, generalist, P/E/R=4o/5/5 | Shared wins | 61405.6 | 180870.26 | 119464.66 | 194.55 |

| Benchmark | Compared pair | Scope | ReAct avg total | Cyclic avg total | Excess (C−R) | Rel. diff. (%) |
|---|---|---|---|---|---|---|
| | | All wins | 61405.6 | 180870.26 | 119464.66 | 194.55 |
| TextCraft-4 | ReAct (4o-mini) vs Spec-Cyc, generalist, P/E/R=4o/4o/4o | Shared wins | 275556.0 | 423724.0 | 148168.0 | 53.77 |
| | | All wins | 264396.0 | 582049.0 | 317653.0 | 120.14 |
| TextCraft-4 | ReAct (4o-mini) vs Gen-Cyc, generalist, P/E/R=4o/4o/4o | Shared wins | 243655.4 | 558693.6 | 315038.2 | 129.3 |
| | | All wins | 264396.0 | 690016.0 | 425620.0 | 160.98 |
| TextCraft-4 | ReAct (5-mini) vs Spec-Cyc, generalist, P/E/R=4o/5/5 | Shared wins | 122673.18 | 485882.27 | 363209.09 | 296.08 |
| | | All wins | 122673.18 | 485882.27 | 363209.09 | 296.08 |
| TextCraft-4 | ReAct (5-mini) vs Gen-Cyc, generalist, P/E/R=4o/5/5 | Shared wins | 122673.18 | 344174.82 | 221501.64 | 180.56 |
| | | All wins | 122673.18 | 344174.82 | 221501.64 | 180.56 |
| ALFWorld | ReAct (4o-mini) vs Spec-Cyc, generalist, P/E/R=4o/4o/4o | Shared wins | 45454.23 | 177099.37 | 131645.14 | 289.62 |
| | | All wins | 44948.19 | 212515.74 | 167567.55 | 372.8 |
| ALFWorld | ReAct (4o-mini) vs Gen-Cyc, generalist, P/E/R=4o/4o/4o | Shared wins | 45693.79 | 173272.44 | 127578.65 | 279.2 |
| | | All wins | 44948.19 | 228945.04 | 183996.85 | 409.35 |
| ALFWorld | ReAct (5-mini) vs Spec-Cyc, generalist, P/E/R=4o/5/5 | Shared wins | 94583.33 | 174485.97 | 79902.64 | 84.48 |
| | | All wins | 94315.32 | 189729.14 | 95413.82 | 101.16 |
| ALFWorld | ReAct (5-mini) vs Gen-Cyc, generalist, P/E/R=4o/5/5 | Shared wins | 94315.32 | 185859.83 | 91544.51 | 97.06 |
| | | All wins | 94315.32 | 207151.84 | 112836.52 | 119.64 |
| Finance-Agent | ReAct (4o-mini) vs Spec-Cyc, generalist, P/E/R=4o/4o/4o | Shared wins | 97959.4 | 120534.0 | 22574.6 | 23.04 |
| | | All wins | 83090.1 | 134142.86 | 51052.76 | 61.44 |
| Finance-Agent | ReAct (4o-mini) vs Gen-Cyc, generalist, P/E/R=4o/4o/4o | Shared wins | 61555.12 | 140562.12 | 79007.0 | 128.35 |
| | | All wins | 83090.1 | 148646.0 | 65555.9 | 78.9 |
| Finance-Agent | ReAct (5-mini) vs Spec-Cyc, generalist, P/E/R=4o/5/5 | Shared wins | 157094.82 | 258076.09 | 100981.27 | 64.28 |
| | | All wins | 144014.85 | 275482.0 | 131467.15 | 91.29 |
| Finance-Agent | ReAct (5-mini) vs Gen-Cyc, generalist, P/E/R=4o/5/5 | Shared wins | 146905.5 | 383115.0 | 236209.5 | 160.79 |
| | | All wins | 144014.85 | 394334.5 | 250319.65 | 173.82 |

# D    Key ablation tables

## D.1    DepDAG-NoRetry diagnostic variant

Table 13 reports DepDAG-NoRetry, the dependency-directed DAG variant without same-subtask retry. We include it as a diagnostic variant to isolate the effect of adding local retry to dependency-directed execution. The main comparison uses DepDAG, which includes local same-subtask retry by default.

Table 13: DepDAG-NoRetry diagnostic variant without same-subtask retry. This appendix table isolates the effect of local retry; the main comparison uses DepDAG with local retry.

| Benchmark | ReAct | | | DepDAG-NoRetry | | | Spec-Cyc | | | Gen-Cyc | | |
|---|---|---|---|---|---|---|---|---|---|---|---|---|
| | SR (%) | TC | STR | SR (%) | TC | STR | SR (%) | TC | STR | SR (%) | TC | STR |
| TextCraft-2 | **94.3% ± 0.2%** | $7.4 \pm 4.3$ | 12.7 | $58.6\% \pm 5.4\%$ | $13.2 \pm 6.1$ | 4.4 | $85.2\% \pm 0.8\%$ | $11.3 \pm 6.5$ | 7.5 | $93.9\% \pm 1.3\%$ | $9.4 \pm 5.6$ | 10.0 |
| TextCraft-3 | **82.5% ± 2.1%** | $16.0 \pm 8.8$ | 5.2 | $17.9\% \pm 9.0$ | $15.4 \pm 6.6$ | 1.2 | $50.4\% \pm 2.3\%$ | $21.6 \pm 11.8$ | 2.3 | $71.5\% \pm 1.5\%$ | $22.1 \pm 11.9$ | 3.2 |
| TextCraft-4 | **45.5% ± 14.8%** | $23.4 \pm 8.4$ | 1.9 | $0.0\% \pm 0.0\%$ | D/A | D/A | $15.2\% \pm 8.6\%$ | $68.8 \pm 20.1$ | 0.2 | $36.4\% \pm 0.0\%$ | $69.1 \pm 15.0$ | 0.5 |
| ALFWorld | $33.8\% \pm 0.4\%$ | $12.9 \pm 7.5$ | 2.6 | $52.2\% \pm 1.6\%$ | $28.1 \pm 14.4$ | 1.9 | **58.2% ± 4.6%** | $30.2 \pm 18.5$ | 1.9 | $56.7\% \pm 3.2\%$ | $28.7 \pm 19.9$ | 2.0 |
| Finance-Agent | $12.4\% \pm 1.3\%$ | $3.0 \pm 0.9$ | 4.1 | $9.5\% \pm 1.3\%$ | $4.5 \pm 2.2$ | 2.1 | $14.3\% \pm 0.0\%$ | $5.3 \pm 2.2$ | 2.7 | **15.2% ± 2.7%** | $6.0 \pm 2.5$ | 2.5 |

## D.2 DepDAG transition audit

To verify that DepDAG does not become a hidden cyclic controller, we inspected completed run artifacts that record typed transitions. The audit covers 47 completed DepDAG runs and 1,419 test cases. Across 7,280 typed transitions, all non-terminal transitions were either same-subtask retries or DAG-forward transitions, and the invalid-transition count was zero.

Table 14: Transition audit for DepDAG from completed run artifacts.

| Quantity | Count |
|---|---|
| Completed DepDAG runs inspected | 47 |
| Test cases inspected | 1,419 |
| Typed transitions inspected | 7,280 |
| Same-subtask retry transitions | 3,511 |
| DAG-forward transitions | 2,722 |
| Terminal transitions | 1,047 |
| Other transition types | 0 |
| Invalid transitions | 0 |

Table 15: Planner–executor–router model-tier sweep with $n$-shot successful-trajectory summaries and generalist executors. We compare Spec-Cyc vs. Gen-Cyc across role allocations and report SR, TC (on successes), STR, and subtask-visitation metrics (ATS/AUS/ASR).

| Benchmark | Graph | Planner | Executor | Router | SR (%) | TC | STR | ATS | AUS | ASR |
|---|---|---|---|---|---|---|---|---|---|---|
| TextCraft-2 | Spec-Cyc | 5-mini | 4o-mini | 4o-mini | $94.9\% \pm 1.0\%$ | $10.3 \pm 5.6$ | 9.2 | $1.8 \pm 1.4$ | $1.5 \pm 1.4$ | $0.3 \pm 0.6$ |
| TextCraft-2 | Spec-Cyc | 4o-mini | 5-mini | 5-mini | $92.5\% \pm 2.0\%$ | $10.9 \pm 4.9$ | 8.5 | $3.5 \pm 2.2$ | $2.5 \pm 1.0$ | $1.0 \pm 1.9$ |
| TextCraft-2 | Spec-Cyc | 4o-mini | 4o-mini | 4o-mini | $83.6\% \pm 1.3\%$ | $12.0 \pm 6.7$ | 7.0 | $2.5 \pm 1.8$ | $1.8 \pm 1.1$ | $0.7 \pm 1.1$ |
| TextCraft-2 | Gen-Cyc | 5-mini | 4o-mini | 4o-mini | $94.4\% \pm 2.2\%$ | $9.4 \pm 5.8$ | 10.0 | $1.20 \pm 1.3$ | $1.0 \pm 1.0$ | $0.2 \pm 0.5$ |
| TextCraft-2 | Gen-Cyc | 4o-mini | 5-mini | 5-mini | $100\% \pm 0.0\%$ | $9.9 \pm 3.1$ | 10.1 | $2.3 \pm 3.3$ | $1.0 \pm 1.1$ | $1.3 \pm 2.7$ |
| TextCraft-2 | Gen-Cyc | 4o-mini | 5-mini | 4o-mini | $99.7\% \pm 0.5\%$ | $11.1 \pm 3.8$ | 9.0 | $6.9 \pm 1.7$ | $3.6 \pm 0.6$ | $3.3 \pm 1.5$ |
| TextCraft-2 | Gen-Cyc | 4o-mini | 4o-mini | 5-mini | $98.9\% \pm 0.2\%$ | $7.9 \pm 5.2$ | 12.5 | $1.9 \pm 1.2$ | $1.7 \pm 0.9$ | $0.2 \pm 0.5$ |
| TextCraft-2 | Gen-Cyc | 4o-mini | 4o-mini | 4o-mini | $93.8\% \pm 0.9\%$ | $9.9 \pm 5.7$ | 9.5 | $1.3 \pm 1.2$ | $1.1 \pm 0.9$ | $0.2 \pm 0.5$ |
| TextCraft-3 | Spec-Cyc | 5-mini | 4o-mini | 4o-mini | $72.4\% \pm 2.5\%$ | $23.2 \pm 12.1$ | 3.1 | $4.5 \pm 2.5$ | $2.9 \pm 1.2$ | $1.6 \pm 1.9$ |
| TextCraft-3 | Spec-Cyc | 4o-mini | 5-mini | 5-mini | $79.7\% \pm 2.5\%$ | $22.1 \pm 11.1$ | 3.6 | $7.5 \pm 4.7$ | $3.4 \pm 1.2$ | $4.1 \pm 4.4$ |
| TextCraft-3 | Spec-Cyc | 4o-mini | 4o-mini | 4o-mini | $57.3\% \pm 1.0\%$ | $24.3 \pm 11.2$ | 2.4 | $5.71 \pm 3.0$ | $3.1 \pm 1.2$ | $2.6 \pm 2.4$ |
| TextCraft-3 | Gen-Cyc | 5-mini | 4o-mini | 4o-mini | $68.7 \pm 2.1\%$ | $23.5 \pm 13.5$ | 2.9 | $4.2 \pm 2.9$ | $2.3 \pm 1.0$ | $1.9 \pm 2.3$ |
| TextCraft-3 | Gen-Cyc | 4o-mini | 5-mini | 5-mini | $100\% \pm 0.0\%$ | $16.6 \pm 8.5$ | 6.0 | $6.1 \pm 3.7$ | $3.1 \pm 0.5$ | $3.0 \pm 3.5$ |
| TextCraft-3 | Gen-Cyc | 4o-mini | 5-mini | 4o-mini | $95.5\% \pm 1.5\%$ | $19.3 \pm 8.7$ | 4.9 | $9.7 \pm 4.2$ | $3.8 \pm 0.5$ | $5.9 \pm 4.0$ |
| TextCraft-3 | Gen-Cyc | 4o-mini | 4o-mini | 5-mini | $91.1\% \pm 2.1\%$ | $21.6 \pm 10.0$ | 4.2 | $4.9 \pm 2.1$ | $3.0 \pm 0.7$ | $1.9 \pm 1.9$ |
| TextCraft-3 | Gen-Cyc | 4o-mini | 4o-mini | 4o-mini | $69.5\% \pm 1.0\%$ | $21.8 \pm 12.4$ | 3.2 | $4.6 \pm 3.2$ | $2.4 \pm 1.0$ | $2.2 \pm 2.7$ |
| TextCraft-4 | Spec-Cyc | 5-mini | 4o-mini | 4o-mini | $42.4 \pm 8.6\%$ | $54.1 \pm 19.5$ | 0.8 | $10.7 \pm 4.1$ | $4.4 \pm 1.1$ | $6.3 \pm 3.9$ |
| TextCraft-4 | Spec-Cyc | 4o-mini | 5-mini | 5-mini | $78.8\% \pm 4.3\%$ | $42.5 \pm 22.3$ | 1.9 | $15.8 \pm 11.7$ | $4.0 \pm 1.6$ | $11.8 \pm 11.1$ |

| Benchmark | Graph | Planner | Executor | Router | SR (%) | TC | STR | ATS | AUS | ASR |
|---|---|---|---|---|---|---|---|---|---|---|
| TextCraft-4 | Spec-Cyc | 4o-mini | 4o-mini | 4o-mini | $18.2\% \pm 0.0\%$ | $71.0 \pm 18.6$ | 0.3 | $17.2 \pm 3.6$ | $4.2 \pm 0.7$ | $13.0 \pm 3.7$ |
| TextCraft-4 | Gen-Cyc | 5-mini | 4o-mini | 4o-mini | $45.6 \pm 7.4\%$ | $58.1 \pm 22.5$ | 0.8 | $11.5 \pm 4.9$ | $3.9 \pm 1.2$ | $7.6 \pm 4.2$ |
| TextCraft-4 | Gen-Cyc | 4o-mini | 5-mini | 5-mini | $100\% \pm 0.0\%$ | $29.5 \pm 9.8$ | 3.4 | $12.1 \pm 5.5$ | $3.3 \pm 0.6$ | $8.8 \pm 5.4$ |
| TextCraft-4 | Gen-Cyc | 4o-mini | 5-mini | 4o-mini | $100\% \pm 0.0\%$ | $38.4 \pm 18.3$ | 2.6 | $16.4 \pm 8.9$ | $4.2 \pm 0.7$ | $12.2 \pm 8.6$ |
| TextCraft-4 | Gen-Cyc | 4o-mini | 4o-mini | 5-mini | $84.9\% \pm 8.6\%$ | $44.6 \pm 22.5$ | 1.9 | $9.6 \pm 4.6$ | $3.3 \pm 0.4$ | $6.4 \pm 4.4$ |
| TextCraft-4 | Gen-Cyc | 4o-mini | 4o-mini | 4o-mini | $48.5 \pm 4.3\%$ | $63.2 \pm 23.9$ | 0.8 | $12.3 \pm 4.9$ | $3.8 \pm 0.9$ | $8.5 \pm 4.7$ |
| ALFWorld | Spec-Cyc | 5-mini | 5-mini | 5-mini | $96.3\% \pm 0.6\%$ | $17.6 \pm 13.5$ | 5.5 | $5.2 \pm 3.8$ | $3.6 \pm 0.9$ | $1.7 \pm 3.4$ |
| ALFWorld | Spec-Cyc | 5-mini | 4o-mini | 4o-mini | $76.9\% \pm 2.7\%$ | $29.7 \pm 18.5$ | 2.6 | $2.6 \pm 1.7$ | $2.0 \pm 1.1$ | $0.6 \pm 1.0$ |
| ALFWorld | Spec-Cyc | 4o-mini | 5-mini | 5-mini | $85.8\% \pm 1.6\%$ | $22.0 \pm 16.5$ | 3.9 | $8.4 \pm 8.4$ | $3.6 \pm 1.2$ | $4.8 \pm 8.1$ |
| ALFWorld | Spec-Cyc | 4o-mini | 4o-mini | 4o-mini | $71.9\% \pm 2.1\%$ | $31.0 \pm 18.9$ | 2.3 | $3.1 \pm 3.5$ | $2.1 \pm 1.1$ | $1.0 \pm 3.0$ |
| ALFWorld | Gen-Cyc | 5-mini | 4o-mini | 4o-mini | $62.9\% \pm 1.5\%$ | $28.0 \pm 19.1$ | 2.2 | $1.7 \pm 2.2$ | $1.1 \pm 1.0$ | $0.6 \pm 1.6$ |
| ALFWorld | Gen-Cyc | 4o-mini | 5-mini | 5-mini | $91.5\% \pm 0.4\%$ | $19.6 \pm 14.8$ | 4.7 | $11.8 \pm 9.3$ | $5.1 \pm 1.4$ | $6.6 \pm 8.5$ |
| ALFWorld | Gen-Cyc | 4o-mini | 4o-mini | 4o-mini | $65.7\% \pm 3.7\%$ | $31.5 \pm 20.5$ | 2.1 | $2.8 \pm 3.0$ | $1.9 \pm 1.3$ | $0.9 \pm 2.1$ |
| Finance-Agent | Spec-Cyc | 5-mini | 4o-mini | 4o-mini | $15.2\% \pm 5.4\%$ | $6.5 \pm 4.3$ | 2.3 | $0.5 \pm 0.8$ | $0.5 \pm 0.8$ | $0.0 \pm 0.0$ |
| Finance-Agent | Spec-Cyc | 4o-mini | 5-mini | 5-mini | $30.5\% \pm 3.6\%$ | $7.1 \pm 3.8$ | 4.3 | $2.3 \pm 2.3$ | $1.8 \pm 1.1$ | $0.5 \pm 1.6$ |
| Finance-Agent | Spec-Cyc | 4o-mini | 4o-mini | 4o-mini | $17.1\% \pm 4.7$ | $4.6 \pm 2.3$ | 3.7 | $0.1 \pm 0.2$ | $0.1 \pm 0.2$ | $0.0 \pm 0.0$ |
| Finance-Agent | Gen-Cyc | 5-mini | 4o-mini | 4o-mini | $11.4\% \pm 2.3\%$ | $5.3 \pm 3.5$ | 2.2 | $0.4 \pm 0.9$ | $0.4 \pm 0.9$ | $0.0 \pm 0.0$ |
| Finance-Agent | Gen-Cyc | 4o-mini | 5-mini | 5-mini | $20.0\% \pm 0.0\%$ | $9.2 \pm 4.4$ | 2.2 | $2.6 \pm 1.9$ | $1.9 \pm 1.5$ | $0.7 \pm 1.5$ |
| Finance-Agent | Gen-Cyc | 4o-mini | 4o-mini | 4o-mini | $17.1\% \pm 2.3\%$ | $5.1 \pm 2.6$ | 3.4 | $0.4 \pm 0.6$ | $0.4 \pm 0.6$ | $0.0 \pm 0.0$ |

Table 16: Executor–router model-tier sweep with $n$-shot summaries and tool-restricted specialist executors. We vary router and executor model tiers (planner fixed) and report SR, TC (on successes), STR, and subtask-visitation metrics (ATS/AUS/ASR).

| Benchmark | Graph | LLM Router | LLM Executor | SR (%) | TC | STR | ATS | AUS | ASR |
|---|---|---|---|---|---|---|---|---|---|
| TextCraft-2 | Spec-Cyc | gpt-5-mini | gpt-4o-mini | $49.1\% \pm 2.9\%$ | $21.0 \pm 4.8$ | 2.3 | $4.3 \pm 1.3$ | $2.6 \pm 0.7$ | $1.7 \pm 0.8$ |
| TextCraft-2 | Spec-Cyc | gpt-4o-mini | gpt-5-mini | $54.2\% \pm 1.1\%$ | $23.2 \pm 5.0$ | 2.3 | $6.3 \pm 1.7$ | $3.4 \pm 0.8$ | $2.9 \pm 1.4$ |
| TextCraft-2 | Spec-Cyc | gpt-4o-mini | gpt-4o-mini | $39.7\% \pm 4.2\%$ | $21.9 \pm 5.5$ | 1.8 | $4.8 \pm 1.5$ | $3.0 \pm 0.8$ | $1.8 \pm 1.1$ |
| TextCraft-2 | Gen-Cyc | gpt-5-mini | gpt-4o-mini | $86.7\% \pm 3.6\%$ | $19.6 \pm 4.6$ | 4.4 | $3.6 \pm 0.9$ | $2.3 \pm 0.5$ | $1.3 \pm 0.6$ |
| TextCraft-2 | Gen-Cyc | gpt-4o-mini | gpt-5-mini | $89.5\% \pm 1.5\%$ | $21.1 \pm 4.2$ | 4.6 | $4.8 \pm 1.2$ | $2.7 \pm 0.6$ | $2.2 \pm 0.9$ |
| TextCraft-2 | Gen-Cyc | gpt-4o-mini | gpt-4o-mini | $70.6\% \pm 0.9\%$ | $20.0 \pm 4.8$ | 3.5 | $3.6 \pm 0.9$ | $2.2 \pm 0.4$ | $1.4 \pm 0.7$ |
| TextCraft-3 | Spec-Cyc | gpt-5-mini | gpt-4o-mini | $28.9\% \pm 2.1\%$ | $36.2 \pm 8.7$ | 0.8 | $7.6 \pm 2.1$ | $3.7 \pm 1.1$ | $3.9 \pm 1.5$ |
| TextCraft-3 | Spec-Cyc | gpt-4o-mini | gpt-5-mini | $29.3\% \pm 6.2\%$ | $38.3 \pm 8.2$ | 0.8 | $10.1 \pm 2.4$ | $4.3 \pm 1.2$ | $5.8 \pm 2.0$ |
| TextCraft-3 | Spec-Cyc | gpt-4o-mini | gpt-4o-mini | $17.1\% \pm 3.4\%$ | $35.7 \pm 7.8$ | 0.5 | $7.7 \pm 1.9$ | $3.7 \pm 1.1$ | $4.0 \pm 1.4$ |
| TextCraft-3 | Gen-Cyc | gpt-5-mini | gpt-4o-mini | $72.8\% \pm 4.5\%$ | $33.9 \pm 8.3$ | 2.1 | $6.2 \pm 1.6$ | $2.8 \pm 0.7$ | $3.4 \pm 1.4$ |
| TextCraft-3 | Gen-Cyc | gpt-4o-mini | gpt-5-mini | $72.8\% \pm 3.2\%$ | $34.5 \pm 8.5$ | 2.1 | $8.1 \pm 2.5$ | $3.1 \pm 0.7$ | $5.0 \pm 2.3$ |
| TextCraft-3 | Gen-Cyc | gpt-4o-mini | gpt-4o-mini | $38.6\% \pm 3.2\%$ | $33.8 \pm 9.0$ | 1.1 | $6.1 \pm 1.7$ | $2.7 \pm 0.7$ | $3.4 \pm 1.5$ |
| TextCraft-4 | Spec-Cyc | gpt-5-mini | gpt-4o-mini | $27.3\% \pm 7.4\%$ | $68.0 \pm 20.0$ | 0.4 | $13.6 \pm 5.0$ | $3.9 \pm 0.9$ | $9.7 \pm 4.7$ |
| TextCraft-4 | Spec-Cyc | gpt-4o-mini | gpt-5-mini | $27.3\% \pm 7.4\%$ | $79.3 \pm 20.4$ | 0.3 | $22.0 \pm 7.0$ | $4.4 \pm 1.2$ | $17.6 \pm 7.0$ |
| TextCraft-4 | Spec-Cyc | gpt-4o-mini | gpt-4o-mini | $0.0\% \pm 0.0\%$ | D/A | D/A | D/A | D/A | D/A |
| TextCraft-4 | Gen-Cyc | gpt-5-mini | gpt-4o-mini | $63.6\% \pm 0.0\%$ | $57.3 \pm 22.1$ | 1.1 | $10.3 \pm 4.1$ | $3.1 \pm 0.8$ | $7.3 \pm 3.7$ |
| TextCraft-4 | Gen-Cyc | gpt-4o-mini | gpt-5-mini | $66.7\% \pm 11.3\%$ | $57.6 \pm 17.6$ | 1.2 | $13.5 \pm 4.8$ | $3.4 \pm 0.7$ | $10.1 \pm 4.3$ |
| TextCraft-4 | Gen-Cyc | gpt-4o-mini | gpt-4o-mini | $15.2\% \pm 4.2\%$ | $84.6 \pm 8.2$ | 0.2 | $27.0 \pm 2.8$ | $5.0 \pm 0.0$ | $22.0 \pm 2.8$ |
| Finance-Agent | Spec-Cyc | gpt-5-mini | gpt-4o-mini | $3.8\% \pm 1.3\%$ | $4.5 \pm 4.3$ | 0.8 | $1.5 \pm 0.9$ | $1.3 \pm 0.4$ | $0.3 \pm 0.4$ |
| Finance-Agent | Spec-Cyc | gpt-4o-mini | gpt-5-mini | $3.8\% \pm 1.3\%$ | $8.0 \pm 6.4$ | 0.6 | $2.0 \pm 1.2$ | $1.5 \pm 0.5$ | $0.5 \pm 0.9$ |
| Finance-Agent | Spec-Cyc | gpt-4o-mini | gpt-4o-mini | $5.7\% \pm 4.0\%$ | $4.7 \pm 2.4$ | 1.2 | $1.5 \pm 0.5$ | $1.5 \pm 0.5$ | $0.0 \pm 0.0$ |
| Finance-Agent | Gen-Cyc | gpt-5-mini | gpt-4o-mini | $16.2\% \pm 3.6\%$ | $3.3 \pm 3.6$ | 4.9 | $1.2 \pm 0.7$ | $1.1 \pm 0.2$ | $0.2 \pm 0.5$ |

| Benchmark | Graph | LLM Router | LLM Executor | SR (%) | TC | STR | ATS | AUS | ASR |
|---|---|---|---|---|---|---|---|---|---|
| Finance-Agent | Gen-Cyc | gpt-4o-mini | gpt-5-mini | $14.3\% \pm 2.3\%$ | $7.2 \pm 4.5$ | 2.0 | $2.5 \pm 1.7$ | $1.5 \pm 0.7$ | $1.0 \pm 1.1$ |
| Finance-Agent | Gen-Cyc | gpt-4o-mini | gpt-4o-mini | $10.5\% \pm 1.3\%$ | $2.0 \pm 0.0$ | 5.3 | $1.0 \pm 0.0$ | $1.0 \pm 0.0$ | $0.0 \pm 0.0$ |

Table 17: Robustness under control-flow fault injection for specialized executors (all components gpt-4o-mini). We compare unperturbed routing to perturbed routing (random subtask redirection) and report SR/TC along with absolute and relative performance deltas.

| Benchmark | Graph | Unperturbed | | Perturbed | | Δ (Pert - Unpert) | | Rel. Diff. |
|---|---|---|---|---|---|---|---|---|
| | | SR (%) | TC | SR (%) | TC | SR (pp) | TC | |
| TextCraft-2 | Spec-Cyc | $39.7\% \pm 4.2\%$ | $21.9 \pm 5.5$ | $41.7\% \pm 2.5\%$ | $20.9 \pm 6.1$ | $2.0\%$ | $-1.0$ | $4.8\%$ |
| | Gen-Cyc | $70.6\% \pm 0.9\%$ | $20.0 \pm 4.8$ | $64.0\% \pm 1.8\%$ | $17.8 \pm 5.9$ | $-6.6\%$ | $-2.2$ | $9.3\%$ |
| TextCraft-3 | Spec-Cyc | $17.1\% \pm 3.4\%$ | $35.7 \pm 7.8$ | $16.7\% \pm 3.2\%$ | $37.5 \pm 9.6$ | $-0.4\%$ | $+1.8$ | $2.3\%$ |
| | Gen-Cyc | $38.6\% \pm 3.2\%$ | $33.8 \pm 9.0$ | $35.0\% \pm 4.1\%$ | $35.5 \pm 10.1$ | $-3.6\%$ | $+1.7$ | $9.3\%$ |
| TextCraft-4 | Spec-Cyc | $0.0\% \pm 0.0\%$ | D/A | $0.0\% \pm 0.0\%$ | D/A | $0\%$ | D/A | D/A |
| | Gen-Cyc | $15.2\% \pm 4.2\%$ | $84.6 \pm 8.2$ | $9.1\% \pm 7.4\%$ | $77.7 \pm 5.4$ | $-6.1\%$ | $-6.9$ | $40.1\%$ |

Table 18: Robustness under control-flow fault injection for generalist executors (all components gpt-4o-mini). We compare unperturbed vs. perturbed routing and report SR/TC along with absolute and relative deltas, characterizing recovery under disrupted control flow.

| Benchmark | Graph | Unperturbed | | Perturbed | | Δ (Pert - Unpert) | | Rel. Diff. |
|---|---|---|---|---|---|---|---|---|
| | | SR (%) | TC | SR (%) | TC | SR (pp) | TC | |
| TextCraft-2 | Spec-Cyc | $83.6\% \pm 1.3\%$ | $12.0 \pm 6.7$ | $49.4\% \pm 0.5\%$ | $19.5 \pm 6.6$ | $-34.2$ | $+7.5$ | $40.9\%$ |
| | Gen-Cyc | $93.8\% \pm 0.9\%$ | $9.9 \pm 5.7$ | $90.1\% \pm 0.7\%$ | $11.1 \pm 5.6$ | $-3.7\%$ | $+1.2$ | $3.9\%$ |
| TextCraft-3 | Spec-Cyc | $57.3\% \pm 1.0\%$ | $24.3 \pm 11.2$ | $19.5\% \pm 4.6\%$ | $36.6 \pm 9.6$ | $-37.8\%$ | $+12.3$ | $66.0\%$ |
| | Gen-Cyc | $69.5\% \pm 1.0\%$ | $21.8 \pm 12.4$ | $65.4\% \pm 2.3\%$ | $25.2 \pm 11.9$ | $-4.1\%$ | $+3.4$ | $5.9\%$ |
| TextCraft-4 | Spec-Cyc | $18.2\% \pm 0.0\%$ | $71.0 \pm 18.6$ | $0.0\% \pm 0.0\%$ | D/A | $-18.2\%$ | D/A | D/A |
| | Gen-Cyc | $48.5\% \pm 4.3\%$ | $63.2 \pm 23.9$ | $42.4\% \pm 4.3\%$ | $57.0 \pm 24.0$ | $-6.1\%$ | $-6.2$ | $12.6\%$ |
| ALFWorld | Spec-Cyc | $71.9\% \pm 2.1\%$ | $31.0 \pm 18.9$ | $62.2\% \pm 0.4\%$ | $29.5 \pm 18.7$ | $-9.7\%$ | $-1.5$ | $13.5\%$ |
| | Gen-Cyc | $65.7\% \pm 3.7\%$ | $31.5 \pm 20.5$ | $50.5\% \pm 2.0\%$ | $29.9 \pm 20.1$ | $-15.2\%$ | $-1.6$ | $23.1\%$ |

Table 19: Unique transition counts (UT) for the generalist sweep with $n$-shot summaries. For each configuration, we report UT over all runs, UT restricted to successful episodes, and UT that never appears in any successful episode (indicative of transitions associated with failure modes or unproductive exploration).

| Benchmark | Graph | Planner | Executor | Router | SR (%) | UT (all) | UT (won) | UT (never won) |
|---|---|---|---|---|---|---|---|---|
| TextCraft-2 | Gen-Cyc | 5-mini | 4o-mini | 4o-mini | $94.4\% \pm 2.2\%$ | 30 | 27 | 3 |
| TextCraft-2 | Gen-Cyc | 4o-mini | 5-mini | 5-mini | $100\% \pm 0.0\%$ | 38 | 38 | 0 |
| TextCraft-2 | Gen-Cyc | 4o-mini | 5-mini | 4o-mini | $99.7\% \pm 0.5\%$ | 25 | 23 | 2 |
| TextCraft-2 | Gen-Cyc | 4o-mini | 4o-mini | 5-mini | $98.9\% \pm 0.2\%$ | 22 | 18 | 4 |
| TextCraft-2 | Gen-Cyc | 4o-mini | 4o-mini | 4o-mini | $93.8\% \pm 0.9\%$ | 22 | 20 | 2 |
| TextCraft-3 | Gen-Cyc | 5-mini | 4o-mini | 4o-mini | $68.7 \pm 2.1\%$ | 31 | 26 | 5 |
| TextCraft-3 | Gen-Cyc | 4o-mini | 5-mini | 5-mini | $100\% \pm 0.0\%$ | 20 | 20 | 0 |
| TextCraft-3 | Gen-Cyc | 4o-mini | 5-mini | 4o-mini | $95.5\% \pm 1.5\%$ | 26 | 25 | 1 |

| Benchmark | Graph | Planner | Executor | Router | SR (%) | UT (all) | UT (won) | UT (never won) |
|---|---|---|---|---|---|---|---|---|
| TextCraft-3 | Gen-Cyc | 4o-mini | 4o-mini | 5-mini | $91.1\% \pm 2.1\%$ | 21 | 20 | 1 |
| TextCraft-3 | Gen-Cyc | 4o-mini | 4o-mini | 4o-mini | $69.5\% \pm 1.0\%$ | 20 | 18 | 2 |
| TextCraft-4 | Gen-Cyc | 5-mini | 4o-mini | 4o-mini | $45.6 \pm 7.4\%$ | 26 | 23 | 3 |
| TextCraft-4 | Gen-Cyc | 4o-mini | 5-mini | 5-mini | $100\% \pm 0.0\%$ | 19 | 19 | 0 |
| TextCraft-4 | Gen-Cyc | 4o-mini | 5-mini | 4o-mini | $100\% \pm 0.0\%$ | 21 | 21 | 0 |
| TextCraft-4 | Gen-Cyc | 4o-mini | 4o-mini | 5-mini | $84.9\% \pm 8.6\%$ | 18 | 16 | 2 |
| TextCraft-4 | Gen-Cyc | 4o-mini | 4o-mini | 4o-mini | $48.5 \pm 4.3\%$ | 33 | 26 | 7 |
| ALFWorld | Gen-Cyc | 5-mini | 4o-mini | 4o-mini | $62.9\% \pm 1.5\%$ | 35 | 30 | 5 |
| ALFWorld | Gen-Cyc | 4o-mini | 5-mini | 5-mini | $91.5\% \pm 0.4\%$ | 77 | 72 | 5 |
| ALFWorld | Gen-Cyc | 4o-mini | 4o-mini | 4o-mini | $65.7\% \pm 3.7\%$ | 69 | 59 | 10 |
| Finance-Agent | Gen-Cyc | 5-mini | 4o-mini | 4o-mini | $11.4\% \pm 2.3\%$ | 13 | 5 | 8 |
| Finance-Agent | Gen-Cyc | 4o-mini | 5-mini | 5-mini | $20.0\% \pm 0.0\%$ | 21 | 15 | 6 |
| Finance-Agent | Gen-Cyc | 4o-mini | 4o-mini | 4o-mini | $17.1\% \pm 2.3\%$ | 12 | 3 | 9 |

Table 20: Unique transition counts (UT) for the specialized-executor sweep with $n$-shot summaries. We report UT over all runs, UT in successful episodes, and UT never observed in successful episodes, enabling comparison of transition diversity and "never-won" behaviors under tool restriction.

| Benchmark | Graph | LLM Router | LLM Executor | SR (%) | UT (all) | UT (won) | UT (never won) |
|---|---|---|---|---|---|---|---|
| TextCraft-2 | Gen-Cyc | gpt-5-mini | gpt-4o-mini | $86.7\% \pm 3.6\%$ | 24 | 22 | 2 |
| TextCraft-2 | Gen-Cyc | gpt-4o-mini | gpt-5-mini | $89.5\% \pm 1.5\%$ | 24 | 22 | 2 |
| TextCraft-2 | Gen-Cyc | gpt-4o-mini | gpt-4o-mini | $70.6\% \pm 0.9\%$ | 25 | 24 | 1 |
| TextCraft-3 | Gen-Cyc | gpt-5-mini | gpt-4o-mini | $72.8\% \pm 4.5\%$ | 22 | 21 | 1 |
| TextCraft-3 | Gen-Cyc | gpt-4o-mini | gpt-5-mini | $72.8\% \pm 3.2\%$ | 25 | 25 | 0 |
| TextCraft-3 | Gen-Cyc | gpt-4o-mini | gpt-4o-mini | $38.6\% \pm 3.2\%$ | 25 | 23 | 2 |
| TextCraft-4 | Gen-Cyc | gpt-5-mini | gpt-4o-mini | $63.6\% \pm 0.0\%$ | 24 | 21 | 3 |
| TextCraft-4 | Gen-Cyc | gpt-4o-mini | gpt-5-mini | $66.7\% \pm 11.3\%$ | 22 | 21 | 1 |
| TextCraft-4 | Gen-Cyc | gpt-4o-mini | gpt-4o-mini | $15.2\% \pm 4.2\%$ | 22 | 17 | 5 |
| Finance-Agent | Gen-Cyc | gpt-5-mini | gpt-4o-mini | $16.2\% \pm 3.6\%$ | 4 | 3 | 1 |
| Finance-Agent | Gen-Cyc | gpt-4o-mini | gpt-5-mini | $14.3\% \pm 2.3\%$ | 4 | 2 | 2 |
| Finance-Agent | Gen-Cyc | gpt-4o-mini | gpt-4o-mini | $10.5\% \pm 1.3\%$ | 7 | 2 | 5 |

Table 21: Observed Gen-Cyc graph size by benchmark. Reported ranges summarize the extracted benchmark-generic graphs across observed configurations. Criterion length is measured in words per edge-transition criterion.

| Benchmark | Average Nodes (observed range) | Avg. criterion length (words) |
|---|---|---|
| ALFWorld | 6–9 | 7.2–30.2 |
| Finance-Agent | 3–7 | 8.3–28.6 |
| TextCraft-2 | 5.5–7 | 9.2–24.1 |
| TextCraft-3 | 6–7 | 9.2–24.1 |
| TextCraft-4 | 5 | 11.0 |

Table 22: Observed Gen-Cyc routing behavior by benchmark. Values are averaged per test case and summarized as ranges across observed Gen-Cyc configurations. Self-loops correspond to repeating the current subtask; inter-node transitions correspond to routing to a different subtask.

| Benchmark | Avg. self-loops / test case | Avg. inter-node transitions / test case |
|---|---|---|
| ALFWorld | 1.0–3.3 | 1.5–11.5 |
| Finance-Agent | 0.0–1.6 | 0.2–3.6 |
| TextCraft-2 | 0.2–3.8 | 1.1–3.1 |
| TextCraft-3 | 2.1–6.4 | 2.6–3.9 |
| TextCraft-4 | 4.4–11.9 | 5.2–7.7 |

Table 23: Observed Spec-Cyc graph size by benchmark. Reported ranges summarize the extracted task-specific graphs across observed configurations. Criterion length is measured in words per edge-transition criterion.

| Benchmark | Average Nodes (observed range) | Avg. criterion length (words) |
|---|---|---|
| ALFWorld | 4.7–5.2 | 8.0–24.0 |
| Finance-Agent | 4.0–5.6 | 12.3–32.7 |
| TextCraft-2 | 3.9–5.1 | 9.4–21.2 |
| TextCraft-3 | 4.7–4.8 | 8.8–8.9 |
| TextCraft-4 | 5.0–6.4 | 8.7–22.1 |

Table 24: Observed Spec-Cyc routing behavior by benchmark. Values are averaged per test case and summarized as ranges across extracted Spec-Cyc configurations with complete transition statistics. Self-loops correspond to repeating the current subtask; inter-node transitions correspond to routing to a different subtask.

| Benchmark | Avg. self-loops / test case | Avg. inter-node transitions / test case |
|---|---|---|
| ALFWorld | 0.9–2.9 | 1.4–9.4 |
| Finance-Agent | 0.0–1.5 | 0.2–2.2 |
| TextCraft-2 | 0.3–0.7 | 1.6–3.5 |
| TextCraft-3 | 2.7–2.8 | 5.2–7.4 |
| TextCraft-4 | 6.3–11.0 | 10.6–14.5 |

# E   Graph-generation stability and reuse

We separate graph construction regimes to clarify what is reused and what is regenerated. Gen-Cyc is generated once per benchmark family and reused across held-out instances within the corresponding configuration. Spec-Cyc is generated per task instance and is therefore expected to vary with instance wording and task-specific entities. DepDAG is also generated per task instance, but its runtime transition set is restricted to the current subtask, DAG-allowed successors, and termination.

Table 25: Graph generation and stability summary. Node-count and criterion-length ranges are reported in Tables 21 and 23; routing behavior ranges are reported in Tables 22 and 24.

| Regime | Generated per instance? | Reused across held-out tasks? | Runtime candidate set | Stability interpretation |
|---|---|---|---|---|
| Spec-Cyc | Yes | No | All subtasks | Highest expressivity and highest graph variability; useful for task-specific recovery but can produce brittle high-precision criteria. |
| Gen-Cyc | No, once per benchmark family | Yes | All subtasks | More stable and reusable; lower instance specificity can regularize routing and reduce brittle criteria. |
| DepDAG | Yes | No | Current subtask, DAG successors, termination | Graph may vary by task, but runtime transitions remain locally retrying or DAG-forward; no arbitrary cross-subtask revisitation. |
| ReAct | No explicit graph | D/A | Tool/action choice only | No explicit subtask graph; workflow is implicit in the single-agent prompt state. |

# F   Success-conditioned subtask visitation

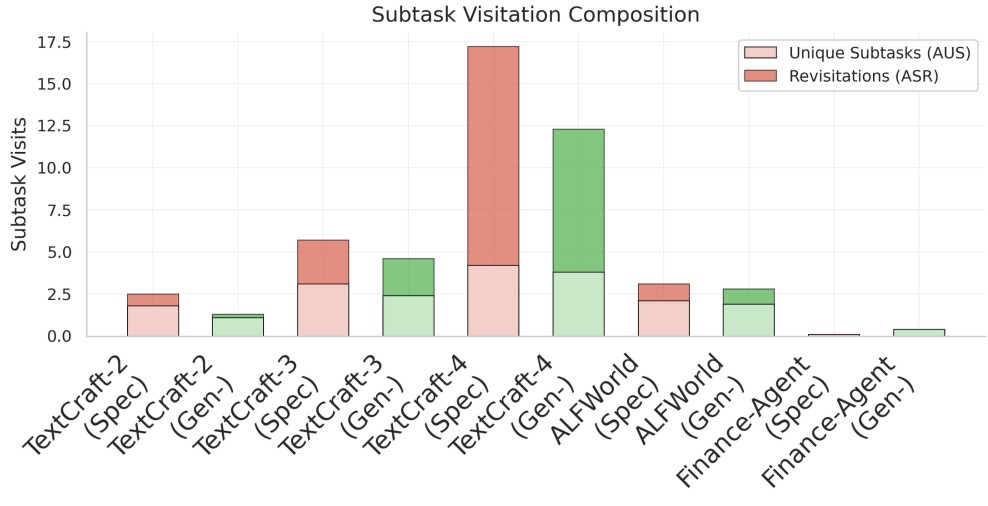

Figure 3: Success-conditioned subtask visitation decomposition. AUS is the number of unique subtasks visited, and ASR is revisitation beyond those first visits. Because the analysis is conditioned on successful episodes, it does not by itself separate productive recovery from unnecessary detours.

## G Environment-level exploration on ALFWorld

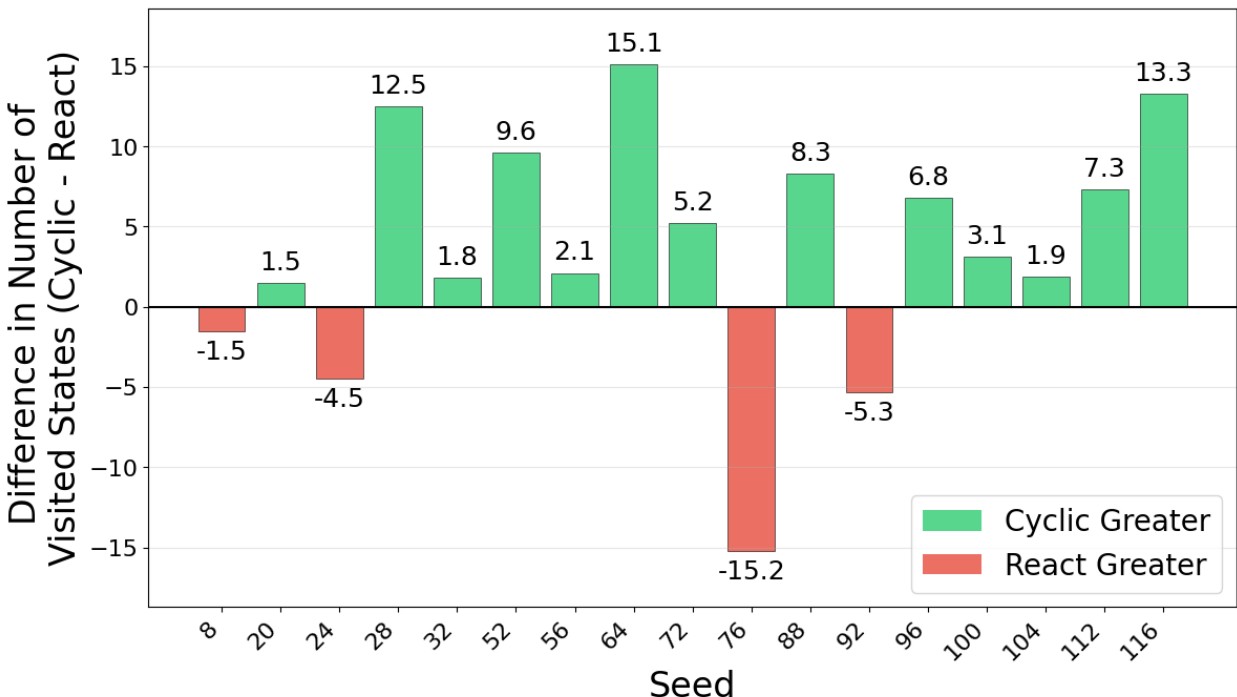

Figure 4: ALFWorld environment-level exploration study: difference in unique environment states visited per episode (cyclic minus ReAct) across the 30-task subsample.

## H    Routing complexity

Routing requires at most $O(|T|)$ criterion-evaluation work per subtask segment, assuming the router state is truncated/summarized to a bounded context window. Let $T = \{t_1, \ldots, t_n\}$ with $n = |T|$. At the end of segment $m$, the router is at node $t_{i_m}$ and considers an admissible set of next-node indices $\mathcal{A}_m \subseteq \{1, \ldots, n\}$. The routing decision is made by evaluating the outgoing transition criteria

$$\{c_{i_m \to j}\}_{j \in \mathcal{A}_m}, \tag{10}$$

i.e., one criterion per admissible candidate $j$.

**Counting argument.**    Define the routing work at segment $m$ as the number of criterion evaluations performed:

$$W_m := |\mathcal{A}_m|. \tag{11}$$

Since $\mathcal{A}_m \subseteq \{1, \ldots, n\}$, it follows immediately that

$$W_m = |\mathcal{A}_m| \le n = |T|. \tag{12}$$

Therefore, the number of criterion evaluations per segment is at most $|T|$, i.e.,

$$W_m \in O(|T|). \tag{13}$$

**Relating evaluations to LLM cost.**    Let $C_{\text{eval}}(m)$ denote the cost of evaluating a single criterion $c_{i_m \to j}$ against the router memory (e.g., measured in tokens processed or bounded wall-clock time under fixed generation limits). If each criterion is evaluated independently, then the routing cost at segment $m$ satisfies

$$C_{\text{route}}(m) = |\mathcal{A}_m| \cdot C_{\text{eval}}(m) \le |T| \cdot C_{\text{eval}}(m). \tag{14}$$

If $C_{\text{eval}}(m)$ is bounded by a fixed prompt/template and fixed decoding limits, then $C_{\text{route}}(m) \in O(|T|)$ as a function of $|T|$.

**One-call implementation.** If routing is implemented as a single LLM call that reads all criteria $\{c_{i_m \to j}\}_{j \in \mathcal{A}_m}$ at once and outputs the next index, then the number of LLM calls per segment is 1, but the input length grows linearly with $|\mathcal{A}_m| \leq |T|$. In this case, the routing cost remains linear in the number of criteria inspected, i.e., $O(|T|)$ in the size of the candidate set.

# I  Failure-mode taxonomy and trace features

To move beyond selected qualitative traces, we define a primary failure-mode taxonomy and extract trace-level features from failed episodes. The current artifact includes deterministic candidate labels and trace packets for manual review; we do not treat the heuristic labels as final human-validated failure counts. Each failed episode can be assigned one primary label based on the earliest dominant cause that prevents task completion; if multiple causes occur, the intended manual label is the one that best explains the unrecovered failure.

Table 26: Primary failure-mode taxonomy used for trajectory analysis.

| Failure mode | Definition |
|---|---|
| Retrieval/source miss | The agent does not retrieve the source, document, filing, page, object, or prerequisite information needed to solve the task. |
| Evidence grounding mismatch | Relevant evidence is retrieved, but the final answer or next action is not grounded in the correct part of the evidence. |
| Synthesis/calculation error | The necessary evidence is present, but the agent combines, compares, calculates, or summarizes it incorrectly. |
| Budget exhaustion before final answer | The trajectory consumes the allowed tool-call budget without submitting or reaching a final answer/goal state. |
| Routing loop / unproductive revisitation | The controller repeatedly revisits subtasks without adding useful state, evidence, or progress. |
| Executor action-selection drift | The executor issues invalid, irrelevant, premature, or goal-misaligned actions despite a plausible high-level subtask. |
| Graph/prompt mismatch | The generated subtask graph or routing criterion misrepresents the task, causing systematic misrouting or impossible subgoals. |
| Other / unresolved | The stored trace does not support a more specific label, or multiple possible causes cannot be separated reliably from the available artifact. |

# J  Qualitative case studies

## J.1  Qualitative routing pathology case study (ALFWorld) on successful trajectory

```
Spec-Cyc trajectory excerpt (ALFWorld, OOD): task = "put a pencil in shelf"; seed=14; won; steps=2; tool
calls=34.

Key pattern:  successful error correction via revisitation (backtracking to recover misplaced object).

Segment 1:  Locate_pencil (misplacement → backtrack)
go desk 1 → see {...  pencil 1}; take pencil 1
error: move pencil 1 to desk 1 (instead of shelf)
go shelf 1 → see {keychain 2} (pencil absent)
go desk 1 → pencil 1 visible again
go shelf 1 → ...
take keychain 2 from shelf 1; distractor: move keychain 2 to shelf 1
go desk 1 → see pencil 1; take pencil 1
repeat error: move pencil 1 to desk 1 ...
... (repeats desk↔shelf to re-acquire pencil after each misplacement) ...
Router analysis:  identifies the loop (repeatedly returning pencil to desk) and transitions to PLACE_PENCIL.

Segment 2:  Place_pencil (loop detection → targeted revisitation)
go shelf 1 → see {keychain 2}; take keychain 2; move keychain 2 to shelf 1
navigation drift: go shelf 2 → see {alarmclock 1} → go shelf 1
... (repeats keychain take/move and shelf1↔shelf2 oscillation) ...
Router analysis:  detects the distraction loop and revisits LOCATE_PENCIL to re-anchor on the target (desk
1) rather than continuing the oscillation.
terminates successfully within budget.
```

Figure 5: Illustrative Spec-Cyc ALFWorld success showing error correction through revisitation: after misplacing the pencil (moving it back to the desk), the controller backtracks to the prior locating context to re-acquire the target and re-attempt placement, rather than committing to a forward-only plan. All agents here are `gpt-4o-mini` with generalist executors.

## J.2    Qualitative routing pathology case study (ALFWorld) on unsuccessful trajectory

```
Spec-Cyc trajectory excerpt (ALFWorld, OOD, seed=2):  task = "put a clean egg in microwave"; lost; steps=7;
tool calls=75.

Step 1:  Locate_clean_egg
...  go/open cabinets ...
open fridge 1 → see {...  egg 1 ...}
take egg 1; examine → "cold egg 1"
error: move egg 1 to fridge 1 (instead of progressing to microwave)
Router analysis:  "mistakenly moved the egg back to the fridge ...  stuck in a loop ..."
Next subtask → TAKE_CLEAN_EGG.

Step 2:  Take_clean_egg
go microwave 1; go fridge 1; inventory → "carrying:  egg 1"
error: move egg 1 to fridge 1  ...    take egg 1  ...    move egg 1 to fridge 1  ...  (repeats)
Router analysis:  "caught in a repetitive loop of moving egg back to fridge ..."
Next subtask → PLACE_CLEAN_EGG_IN_MICROWAVE.

Step 3:  Place_clean_egg_in_microwave
take egg 1 from fridge 1
error: move egg 1 to fridge 1
go/open microwave 1 → contains apple 1, potato 1
distractor loop: take apple 1; move apple 1 to microwave 1; take potato 1; move potato 1 to microwave 1; ...
(repeats)
Router analysis:  "moved egg back to fridge ...  stuck in a loop ..."
...

Step 5:  Place_clean_egg_in_microwave (later succeeds)
go microwave 1; move egg 1 to microwave 1 (subtask success)
Next subtask → CLEAN_EGG.

Step 6:  Clean_egg
take egg 1 from microwave 1; go sinkbasin 1; clean egg 1 with sinkbasin 1 (confirmed)
error: move egg 1 to sinkbasin 1  ...    take egg 1  ...    move egg 1 to sinkbasin 1  ...  (repeats)
Router analysis:  "cleaning confirmed, but objective still not met ...  examine egg to verify ..."
Next subtask → TAKE_CLEAN_EGG.

Step 7:  Take_clean_egg
examine egg 1 → "cool and clean egg 1"
Router analysis:  "egg confirmed clean, yet objective still not met ..."
Next subtask → PLACE_CLEAN_EGG_IN_MICROWAVE.
```

Figure 6: Illustrative Spec-Cyc ALFWorld failure trace showing oscillation between task-specific nodes (TAKE/PLACE/CLEAN) and feedback-entangled criteria ("objective not met"), consistent with the discussion claim that higher expressivity can increase criterion ambiguity and brittleness. All agents here are `gpt-4o-mini` with generalist executors. This trace also highlights a task-specific ambiguity: node labels encode high-precision predicates (e.g., "LOCATE *CLEAN* EGG") while the environment initially only reports "cold"; this can bias routing toward believing the node intent over the actual observations, yielding premature transitions into "clean-egg" subtasks and brittle oscillation when executor actions contradict the intended predicate.

35

## J.3 Qualitative executor-style pathology case studies (TextCraft-3) on unsuccessful trajectory

```
Gen-Cyc trajectory excerpt (TextCraft-3, specialized executor):  task = "craft polished granite"; seed=85;
lost; steps=10; tool calls=51.

Key pattern:  repeated self-loop on Gather_materials under tool-restricted specialization.
The specialized "raw-materials" executor repeatedly queries for diorite/granite as if they are directly
gatherable, leading to unproductive retries and budget burn rather than progressing to prerequisite
crafting.

Step 1:  Gather_materials
try get diorite → Could not find diorite
try get granite → Could not find granite
get quartz → Got 4 quartz
Router:  "attempted to gather materials but did not collect any" ...  Next → GATHER_MATERIALS (self-loop).

Step 2:  Gather_materials (self-loop)
get granite → Could not find granite
get diorite → Could not find diorite
get quartz → Got 4 quartz
Router:  "has 4 quartz, 0 diorite, 0 granite ...  try gathering diorite again" ...  Next →
GATHER_MATERIALS.

Steps 3-6:  Gather_materials (repeated self-loops)
...  repeated attempts:  get diorite → Could not find diorite; get granite → Could not find granite ...
Router repeatedly concludes:  "still needs 4 diorite ...  attempt to gather again" ...

Step 7:  Gather_materials (partial prerequisite attempt, still fails)
get cobblestone → Got 8 cobblestone
get quartz → Got 4 quartz (now 8 quartz total)
attempt get/craft diorite → Could not find diorite
Router:  "attempted to gather 4 diorite ...  unsuccessful" ...  Next → GATHER_MATERIALS.

Steps 8-10:  Gather_materials (terminal budget burn)
...  repeated Could not find diorite (and occasional Could not find granite) ...
End:  still has quartz (4-8) and cobblestone (8), but never reaches the intermediate crafting needed to
obtain granite and thus polished granite.
```

Figure 7: Illustrative Gen-Cyc TextCraft-3 failure under a specialized raw-materials executor: the controller repeatedly self-loops on GATHER_MATERIALS, issuing direct "get" queries for non-gatherable intermediates (diorite/granite) and consuming budget with retries. This qualitative trace supports the coordination-tax view that tool-restricted specialization can underperform generalists when intermediate prerequisites must be inferred and executed via cross-step coordination. All agents here are gpt-4o-mini.

```
Gen-Cyc trajectory excerpt (TextCraft-3, generalist executor):  task = "craft polished granite"; seed=85;
lost; steps=10; tool calls=50.

Key pattern:  the generalist can craft intermediates, but still fails under action-selection drift and
goal/recipe misalignment.
Unlike the specialized raw-materials executor, the generalist occasionally performs prerequisite crafting
(e.g., craft diorite), but repeatedly issues wrong craft commands (unrelated items) and repeatedly attempts
terminal goals (craft polished_granite) before crafting granite.

Step 1:  Identify_target_item
attempt craft polished_granite → Could not find enough items ...
get quartz → Got 4 quartz; try get diorite → Could not find diorite
Next → ACQUIRE_RAW_MATERIALS.

Step 2:  Acquire_raw_materials
get granite → Could not find granite; get diorite → Could not find diorite
inventory → [quartz] (4); get quartz → Got 4 quartz (now 8); get cobblestone → Got 4 cobblestone
Next → ACQUIRE_RAW_MATERIALS.

Step 3:  Acquire_raw_materials (drift to oversized requests)
...  get granite → Could not find granite; get diorite → Could not find diorite ...
get 100 cobblestone → Got 100 cobblestone; get 100 quartz → Got 100 quartz.
Next → DETERMINE_RAW_MATERIALS.

Steps 4-5:  Determine_raw_materials (still treats intermediates as gatherable)
...  repeated get diorite/get granite → Could not find ...
inventory → large quartz/cobblestone counts.
...

Step 6:  Determine_raw_materials (partial progress, then wrong crafts)
craft 2 diorite using 2 quartz, 2 cobblestone → Crafted 2 diorite
wrong craft: craft dark oak planks → fails
premature terminal craft: craft polished_granite → fails (no granite)
...  get diorite → Could not find diorite; get quartz → Got 2 quartz
Next → ACQUIRE_RAW_MATERIALS.

Step 9:  Determine_raw_materials (repeat:  crafts diorite, skips granite)
inventory → ...  [diorite] (2); craft 2 diorite → Crafted 2 diorite (now 4)
still skips granite: craft polished_granite → fails
Next → ACQUIRE_RAW_MATERIALS.

Step 10:  Acquire_raw_materials (terminal drift persists)
get granite → Could not find granite; inventory → [diorite] (4)
wrong craft (repeats): craft dark oak planks → fails
premature terminal craft (repeats): craft polished_granite → fails
End:  substantial quartz/cobblestone and some diorite, but never crafts granite → cannot craft
polished_granite.
```

Figure 8: Illustrative Gen-Cyc TextCraft-3 failure under a generalist executor. The generalist occasionally makes prerequisite progress (e.g., successfully crafting `diorite`), but still exhibits action-selection drift: it issues unrelated craft commands (e.g., `dark_oak_planks`) and repeatedly attempts the terminal goal (`polished_granite`) without executing the missing intermediate (`granite`). Compared to the specialized trace (Appendix Fig. 7), the failure mode shifts from tool-restriction-induced dead-ends to mis-executed command selection, underscoring that generalist access is necessary but not sufficient when routing/execution is weak. All agents here are `gpt-4o-mini`.

