# OpenReview forum: "Complete Cyclic Subtask Graphs for Tool-Using LLM Agents: Flexibility, Cost, and Bottlenecks in Long-Horizon Workflows"
_TMLR — Under review for TMLR_

### Review · Reviewer_U43p · 2026-05-17

**Summary Of Contributions:**

# Overview

This paper proposes a taxonomy for long horizon tool using tasks, defining four rough types of
tasks. Three of these task types are represented by a benchmark, which is used to compare four different agent control architectures: ReAct allows the LLM to decide the next tool, while the other 3 proposed architectures (Spec-Cyc, Gen-Cyc, and DepDag) use LLM generated task decompositions and to evaluate whether the agent is ready to transition to the next task. Results are analyzed with conclusions drawn about which architectures suit each of the four task types, with commentary on . The paper makes these comparisons using weaker model tiers (gpt-4o-mini and gpt-5-mini) for graph generation/orchestration, arguing that weaker models provide headroom to observe orchestration effects that would be obscured by stronger and larger models.

# Strengths

## Thorough experimental design.

The evaluation spans three distinct benchmarks (TextCraft, ALFWorld, Finance-Agent) rather than a single benchmark, compares multiple control architectures (ReAct, DepDAG, Spec-Cyc, Gen-Cyc).

## Design choices probe agent resilience and adaptation:

Fault-injection technique (Section 4.3) tests how agents recover when the router is forced into wrong choices, and the n-shot summaries mechanism (Section 4.1.6) feeds the planner examples of past successful runs as context - mimicing agent memory.

##  Inclusion of benchmarks with poor performance

Reporting is honest: benchmarks where the proposed methods underperform are included rather than dropped, and Table 8 reports token cost alongside accuracy.

##  Non-obvious findings

The finding that benchmark-generic graphs used with the gen-cyc architecture are often competitive with task-specific ones is non-obvious and has practical implications for establishing representative graphs that are adapted to similar, yet different tasks.

The cost-flexibility tradeoff noted is also real and significant (cyclic methods are 2-5x more expensive in tokens, also increasing latency) - useful for real-time, user facing agents.

# Weaknesses

* The selection matrix recommends architectures for task types based on one benchmark per type; one row has no empirical support.
* LLM hyperparameters are unspecified, including whether reasoning was enabled for gpt-5-mini.
* The DepDAG baseline lacks subtask-level recovery, which is standard in agent-literature baselines (e.g., AdaPT) and supported by the benchmarks' idempotent semantics.
* Failure analysis is illustrative (three traces in Appendix G) rather than systematic; the trajectory data supports decomposition but is not analyzed for it.
* Statistical inference is thin (three seeds, small test sets); several method differences sit within reported standard deviations.
* The Finance-Agent bottleneck claim is unsupported: peak performance is 30%, far below saturation, so the paper's rationale for using weaker tiers does not apply here.
* Model selection is narrow: one architectural family (dense), one model size (mini) from one provider (OpenAI), no open-source or other-provider conditions.

**Audience:**

Yes

**Audience Explanation:**

Yes. Several non-obvious findings are useful for researchers and practitioners building agent systems where reliability and recovery matter (particularly non-coding domains).

1. benchmark-generic graphs are often competitive with task-specific ones (Gen-Cyc result)
2. fault-injection study is a useful stress-testing lens that makes findings more applicable to real world applications
3. the cost-flexibility tradeoff is made empirically visible, and the negative result on TextCraft suggests practitioners carefully decide when to add workflow control.

These address design choices researchers and practitioners currently make without clear empirical guidance.

**Broader Impact Concerns:**

None observed.

**Claims And Evidence:**

Yes

**Claims Explanation:**

In the submitted version, the claims are partially supported. Most of the unsupported claims are addressable through ablation, re-analysis of existing data, or claim reduction.

**Supported:**
Benchmark-level empirical findings (e.g., ReAct outperforms cyclic methods on TextCraft; Gen-Cyc is often competitive with Spec-Cyc) are reasonably supported by the experimental setup.

**Not supported:**
* The task-regime taxonomy and selection matrix (Table 1) recommend architectures for task types based on a single benchmark per type, with one row having no empirical support at all.
*  The Finance-Agent bottleneck claim (that retrieval/grounding dominate over workflow design) is confounded with model capability - peak performance is 30%, well below saturation, so the paper's own rationale for using weaker model tiers does not apply.
*  Cost analysis and model-tier comparisons rely on unspecified hyperparameters (notably reasoning configuration for gpt-5-mini
* the DepDAG baseline lacks recovery semantics on benchmarks that support them, making the cyclic-vs-DAG comparison partly a recovery-vs-no-recovery comparison

**Requested Changes:**

## Requested Changes

### Critical for acceptance

- **Reframe the task-regime taxonomy and selection matrix.** As written, Table 1 recommends architectures for task types based on one benchmark per type, with one row unsupported entirely. Qualify the language to avoid overreaching claims.
- **Specify all LLM hyperparameters per model and role.** Document whether reasoning effort was enabled and if so, to what degree for gpt-5-mini. If no reasoning model was used for paper analysis, there should be at least one included given that default API behaviour for current models (Opus 4.7, GPT-5.5) enable reasoning.
- **Improve the DepDAG architecture to include subtask-level retry.**
- **Add failure-mode analysis of agent trajectories**, classifying primary cause of failure type.
- **Broaden model coverage** include a frontier model to support the retrieval/grounding claim for Finance-Agent, at least one MoE model, and at least one different model provider (Gemini, Anthropic, open source).

### Would strengthen
- Report confidence intervals or significance tests on architectural comparisons, especially for Finance-Agent.
- Report graph stability for architectures where graphs are generated at each agent runtime; note whether deviation in graph compared to baseline affects results.

---

> ### Author Response · Authors · 2026-06-11
>
> We thank the reviewer for the detailed and actionable critique. We agree with the main concerns and revised the paper to reduce overclaiming, strengthen the dependency-directed baseline, document model settings, and add additional diagnostics. For overlapping points on novelty, reproducibility, practical sparse controllers, and robustness limitations, please also see responses [1]-[6] to Reviewer oWhQ. The main changes addressing this review are:
>
> [1] Task-regime taxonomy and selection matrix. We substantially reframed the taxonomy. The Abstract, Introduction, Sec. 3.3, Table 1, Sec. 6, and Conclusion now describe the benchmarks as case studies that suggest workflow signatures, not as exhaustive representatives of task classes. Table 1 is now a qualified diagnostic matrix rather than a universal architecture selector, and the unsupported mixed-enterprise row was removed.
>
> [2] LLM hyperparameters and reasoning settings. We added Sec. 4.1.5 and Appendix B.3/Table 8 to document model roles, temperatures, output caps, provider-default sampling fields, tool-call budgets, judge settings, and reasoning settings. In particular, Table 8 states that gpt-5-mini used low reasoning effort and that gpt-4o-mini had no reasoning-effort setting.
>
> [3] DepDAG with subtask-level retry. We agree that the original comparison partly conflated cyclicity with recovery access. We therefore added DepDAG+Retry as the main dependency-directed baseline in Sec. 3.2.4 and Sec. 4.1.2. Table 2 now reports ReAct, DepDAG+Retry, Spec-Cyc, and Gen-Cyc in the main no-n-shot comparison. The no-retry DepDAG is moved to Appendix D.1, and Appendix D.2 adds a transition audit verifying that DepDAG+Retry only used same-subtask retries, DAG-forward transitions, and termination, with zero invalid transitions in the audited artifacts.
>
> [4] Failure-mode analysis. We added a main-paper failure-mode paragraph after A4, plus Appendix B.6 and Appendix H. The added diagnostics separate retrieval/source misses, grounding mismatches, budget exhaustion, routing loops, executor drift, and graph/prompt mismatch. We are careful not to overstate this addition: the manuscript now calls these trace-derived diagnostic labels rather than fully human-adjudicated failure counts.
>
> [5] Finance-Agent model coverage and bottleneck claim. We agreed that the original Finance-Agent bottleneck claim was too strong. We added Appendix B.5 with model-coverage checks using Gemini 2.5 Pro and DeepSeek-v3.1-Terminus through OpenRouter, and Appendix B.6 with tool-use/failure diagnostics. We also rewrote Sec. 4.4 and Sec. 5 to make the conclusion narrower: Finance-Agent is treated as an evidence-pipeline stress test, and the results indicate that workflow control alone is insufficient under the current tools, budget, and evidence pipeline. We no longer present small controller differences as conclusive architecture rankings.
>
> [6] Statistical uncertainty. We added Appendix B.4 with Wilson 95% confidence intervals for small held-out sets and now explicitly caution against overinterpreting close Finance-Agent differences. This is also reflected in Sec. 4.4 and the revised Discussion.
>
> [7] Graph stability. We added Appendix E to clarify which graphs are generated per instance, which are reused, and what runtime candidate sets are available for Spec-Cyc, Gen-Cyc, DepDAG+Retry, and ReAct. This addresses the concern that generated graph variation could affect results.
>
> [8] n-shot summaries and tool exposure. We clarified n-shot construction and held-out evaluation in Sec. 4.1.6 and Appendix B.2. We also preserved the tool-exposure ablation and clarified why ALFWorld is excluded from that sweep: its interface is monolithic, so any tool partition would be arbitrary and could bias the comparison.
>
> We believe these changes directly address the critical requests while keeping the revised claims appropriately scoped to the evidence.

---

### Review · Reviewer_oWhQ · 2026-05-28

**Summary Of Contributions:**

This paper studies complete cyclic subtask graphs for tool-using LLM agents, where executable subtask nodes are fully connected and a router selects transitions using natural-language criteria. The authors compare task-specific and benchmark-generic cyclic graphs against ReAct and a dependency-directed DAG on TextCraft, ALFWorld, and Finance-Agent, reporting success, tool-call efficiency, revisitation behavior, token cost, and fault-injection robustness. The main strength is a clear empirical framing of the flexibility-cost tradeoff in long-horizon agent workflows. The main weakness is that the algorithmic novelty is modest, and some conclusions, especially the task-regime taxonomy and robustness interpretation, are supported mainly by limited benchmark evidence and coarse episode-level metrics.

**Additional Comments:**

The paper is clear and useful as an empirical study, and I appreciate that the authors do not oversell complete connectivity as universally optimal. My main reservation is that the technical idea is simple, and the strength of the submission depends heavily on the completeness of the empirical comparison. With clearer positioning, stronger practical baselines, and more careful robustness analysis, I would be more confident in acceptance. Overall recommendation: borderline accept / weak accept; confidence: medium.

**Audience:**

Yes

**Audience Explanation:**

Researchers working on LLM agents, planning-execution decomposition, workflow control, and multi-agent orchestration would likely find the findings relevant. The paper is useful because it empirically separates cases where explicit revisitation improves recovery from cases where it mainly increases routing and token overhead. Even if the complete cyclic graph is unlikely to be the best final deployment architecture, the study provides a useful lens for deciding when backtracking, verification, and recovery routes should be made explicit.

**Broader Impact Concerns:**

I do not see major additional broader-impact concerns beyond those already discussed. The paper studies workflow control for tool-using agents, and the main risk is over-reliance on automated completion in high-stakes domains. Since the Finance-Agent results show persistent retrieval, grounding, and synthesis failures, the authors should make clear that cyclic revisitation is not a substitute for evidence verification, domain expertise, or human review in financial, legal, medical, or safety-critical workflows.

**Claims And Evidence:**

Yes

**Claims Explanation:**

The main empirical claims are mostly supported by the reported results: cyclic control helps most clearly on ALFWorld, often adds coordination cost on TextCraft, and gives only small gains on Finance-Agent. The token and tool-call analyses also support the claim that flexibility is costly. However, some claims are somewhat stronger than the evidence warrants. The regime taxonomy is plausible but largely post hoc, the fault-injection study measures episode-level retention rather than fine-grained recovery dynamics, and the paper does not compare against several practical sparse or hand-designed recovery controllers. Overall, the evidence is sufficient for the paper’s diagnostic interpretation, but less convincing for broad design recommendations.

**Requested Changes:**

Critical: the authors should clarify the exact novelty relative to prior graph-based and state-machine agent controllers, and position the main contribution more explicitly as an empirical diagnostic study. They should also add or justify the absence of stronger practical baselines, such as sparsified cyclic graphs, learned transition pruning, or hand-designed recovery state machines.

The paper should provide complete prompts and evaluation scripts, and should report stronger uncertainty analysis where performance gaps are small. The limitations of episode-level fault injection should also be stated more explicitly. Strengthening: report wall-clock latency, expand the quantitative failure analysis beyond selected traces, clarify the use of n-shot summaries across benchmarks, and reduce repeated discussion of the same flexibility-cost tradeoff.

---

> ### Author Response · Authors · 2026-06-11
>
> We thank the reviewer for the careful and constructive assessment. We agree that the paper is strongest when framed as an empirical diagnostic study rather than as a claim that complete connectivity is a generally optimal controller. We revised the manuscript accordingly.
>
> [1] Novelty and positioning. We revised the Abstract, Introduction, Contributions, Related Work, Discussion, and Conclusion to position the contribution as a diagnostic study of unrestricted subtask revisitation. The revised Related Work now explicitly distinguishes our goal from state-machine and graph-optimization systems: prior work typically asks how to construct or optimize a workflow, whereas we isolate one control-flow variable, whether executable subtasks may revisit any other executable subtask, and measure the resulting recovery/cost behavior.
>
> [2] Practical baselines and recovery controllers. We agree that the original dependency-directed baseline was too weak as a recovery comparison. We added DepDAG+Retry as the main dependency-directed controller in Sec. 3.2.4 and Sec. 4.1.2, and report it in the main comparison in Table 2. DepDAG+Retry permits same-subtask retry, DAG-forward transitions, and termination, while still disallowing arbitrary cross-subtask revisitation. The previous no-retry DepDAG is retained only as a diagnostic appendix variant (Appendix D.1). We also added a transition audit (Appendix D.2) showing that DepDAG+Retry did not become a hidden cyclic controller: all audited non-terminal transitions were either same-subtask retries or DAG-forward transitions. In response to the concern about sparse/pruned and hand-designed recovery controllers, we added a Discussion paragraph explaining that these are practical deployment targets, but require extra benchmark-specific design, supervision, or post-hoc transition-utility estimates. We now frame the complete graph as the diagnostic upper-flexibility condition from which such sparse controllers can be distilled.
>
> [3] Prompts, scripts, and reproducibility. We added explicit reproducibility language in Sec. 4.1.5, Sec. 6, and Appendix A, and the supplementary artifact now includes prompt templates, structured-output schemas, execution scripts, result-processing utilities, and anonymized logs. We also added Appendix B.3/Table 8 documenting model roles, decoding settings, output caps, reasoning-effort settings, provider defaults, tool-call budgets, and judge settings.
>
> [4] Uncertainty where gaps are small. We agree that small Finance-Agent gaps should not be overinterpreted. We added Appendix B.4 with Wilson 95% intervals for small held-out sets and rewrote the Finance-Agent claims in Sec. 4.4 and Sec. 5 as sensitivity/contrastive evidence rather than conclusive architecture rankings.
>
> [5] Fault-injection and failure analysis. We revised Sec. 4.3 to clarify that fault injection is a harsh robustness stress test, not a realistic deployment policy, and that the result is episode-level success retention rather than event-level repair latency. We added a main-paper failure-mode paragraph after A4 and added Appendix B.6 and Appendix H for trace-derived Finance-Agent diagnostics and a failure-mode taxonomy. We explicitly state that these labels are diagnostic/heuristic rather than fully human-adjudicated failure counts.
>
> [6] Latency and broader impact. We agree that wall-clock latency is important. Because the stored logs record tokens and tool calls but not wall-clock times, Sec. 4.1.3, Sec. 5, and Sec. 6 now state this limitation and treat tokens/calls as compute proxies. We also strengthened Sec. 7 to state that cyclic revisitation is not a substitute for source verification, domain expertise, or human review in financial, legal, medical, or safety-critical settings.
>
> We hope these changes make the paper more clearly scoped, more reproducible, and more careful about the limits of the empirical evidence.

---

### Review · Reviewer_w6nE · 2026-06-26

**Summary Of Contributions:**

The paper presents a framework for allowing LLM agents to complete tasks by choosing subtasks from a graph. An agent runs on each subtask, while another agent routes to the next choices. In this particular paper, the graph is a complete graph, so retrying prior steps is possible. The authors consider 3 different agentic benchmarks. On each one they consider task-specific and benchmark-specific cyclic graphs, a non-cyclic baseline, and ReAct, a baseline from the literature. They find that on different benchmarks, cyclic revisitation may be very useful, sometimes useful, or actively unhelpful; each of the three benchmarks is essentially a case study for one of those situations.

**Audience:**

Yes

**Audience Explanation:**

The topic is definitely relevant to the TMLR audience.

**Broader Impact Concerns:**

None.

**Claims And Evidence:**

Yes

**Claims Explanation:**

As a self-contained work, the experiments appear to support the empirical claims being made; the authors accurately characterize the performance in their 3 different settings. The settings are well-accepted benchmarks. The baseline appears to be a standard one in the area.

TMLR doesn't consider "novelty" or "interest" as a required criterion. However, there does have to be some separation from prior work. The basic idea of having agents make tool calls following a subgraph that allows loops seems like it must have been done before. While the authors attempt to explain what is different about their approach, the writing about this is far from clear to a non-expert, and in those parts of the text, references to prior methods (e.g. state machines) don't come with citations to the specific prior work that used those methods. I would hope that other reviewers with more background knowledge, or the authors, can make this clearer.

**Requested Changes:**

Figure 2 is completely unreadable, I would recommend finding a different way to present that information.

Please further clarify relation to prior work.